# Towards Robust Parameter-Efficient Fine-Tuning for Federated Learning

**Xiuwen Fang[1], Mang Ye[1,2]\***

[1] School of Computer Science, Wuhan University, Wuhan, China
[2] Taikang Center for Life and Medical Sciences, Wuhan University, Wuhan, China
`{fangxiuwen, yemang}@whu.edu.cn`

## Abstract

Federated Learning enables collaborative training across decentralized edge devices while preserving data privacy. However, fine-tuning large-scale pre-trained models in federated learning is hampered by substantial communication overhead and client resource limitations. Parameter-efficient fine-tuning methods like Low-Rank Adaptation (LoRA) reduce resource demands but suffer from aggregation discrepancies and heightened vulnerability to label noise, particularly in heterogeneous federated settings. In this paper, we introduce RFedLR, a robust federated PEFT framework designed to overcome these challenges. RFedLR integrates two key components: (1) Sensitivity-aware robust tuning, which identifies and selectively updates noise-sensitive parameters to bolster local robustness against label noise, and (2) Adaptive federated LoRA aggregation, which dynamically weights and aggregates LoRA updates based on their importance and stability to minimize bias and noise propagation. Comprehensive experimental validation shows RFedLR outperforms existing methods, achieving superior accuracy and robustness in noisy federated scenarios. Our code is available at: https://github.com/FangXiuwen/RFedLR

## 1 Introduction

The proliferation of edge devices generates massive siloed data volumes, whose direct aggregation and utilization are impeded by significant privacy, regulatory, and transmission constraints [1]. To meet the challenge, Federated Learning (FL) [2] is a distributed machine learning framework that facilitates collaborative model training across multiple decentralized clients without sharing sensitive private data [3–5]. In the classic FedAvg algorithm, clients train models locally on their private datasets and transmit updated parameters to the server for weighted averaging.

Recently, large-scale Pre-Trained Models (PTMs) show impressive performance in various tasks [6]. Combining the knowledge of large models with the distributed nature of FL [7], via client-side fine-tuning and server-side aggregation, offers improved performance and privacy guarantees [8–10]. However, applying Full Fine-Tuning (FFT) to large-scale PTMs in FL faces significant challenges [11]: (1) the substantial communication overhead incurred by transmitting large parameter updates across resource-constrained networks, and (2) the excessive storage and computational demands that exceed the capabilities of typical edge devices. Consequently, to facilitate FL with large-scale models on resource-constrained edge devices, we adopt Low-Rank Adaptation (LoRA) [12], a representative Parameter-Efficient Fine-Tuning (PEFT) technique, as the paradigm for client-side updates. LoRA drastically reduces trainable parameters by decomposing weight updates into low-rank matrices (Fig. 1), making PTMs fine-tuning viable on edge devices.

---

\*Corresponding author.

39th Conference on Neural Information Processing Systems (NeurIPS 2025).

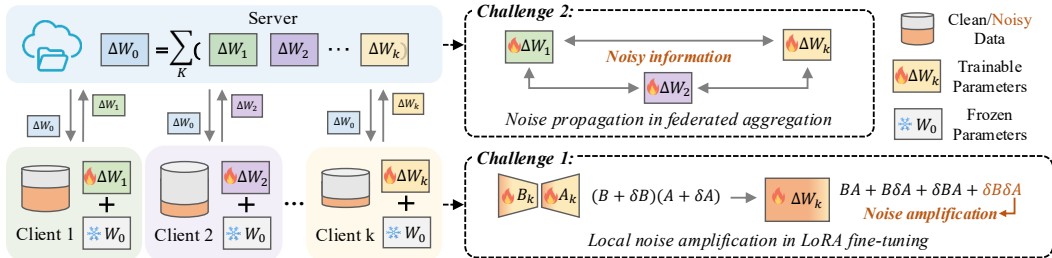

Figure 1: Illustration of federated learning with parameter-efficient fine-tuning, where clients possess noisy datasets with different noise rate. Only a small number of trainable parameters are updated and transferred.

However, integrating LoRA into FL presents distinct challenges. Naive federated averaging of LoRA components, such as FedLR [13] and SLoRA [14], leads to aggregation discrepancies because the average of low-rank decompositions is not equivalent to the decomposition of the average of full updates (Eq. 3). This can result in suboptimal global models. Attempts to mitigate this, FFA-LoRA [15] freezes the non-zero initialized low-rank matrix and only allows the zero-initialized low-rank matrix to be trained and aggregated. However, this rigidity may constrain the model expressiveness. RoLoRA [16] alternately updates between matrices across communication rounds, avoiding permanently freezing either matrix, but it still imposes a rigid update schedule. Critically, existing methods lack sufficient flexibility, potentially leading to the neglect of valuable updates and suboptimal performance in heterogeneous FL [1].

Existing federated PEFT methods are based on the assumption that the clients have clean private datasets. However, the FL system involves numerous distributed clients whose private data quality varies significantly, often including noisy labels. Label noise in FL can stem from diverse sources, such as human expertise limitations, cost compromises, or deliberate noise injection for privacy and fairness. The presence of label noise poses a significant barrier to achieving robust performance in federated PEFT. FFT exhibits resilience to label noise owing to its expansive parameter space.

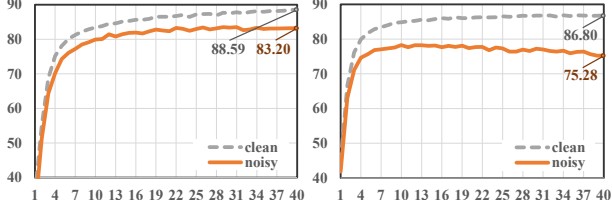

(a) Accuracy of federated FFT  (b) Accuracy of federated LoRA

Figure 2: Comparison of federated fine-tuning methods under clean and noisy (20% pairflip noise) conditions. The average test accuracy (%) of all clients is shown. (a) Federated FFT shows stable performance with minor accuracy loss under noise. (b) Federated LoRA demonstrates greater susceptibility to noise.

The constrained parameterization of LoRA makes it more susceptible to overfitting noisy labels, as illustrated in Fig. 2. Furthermore, the LoRA decomposition itself can exacerbate bias from label noise. Label noise introduces bias terms in the two low-rank decomposition matrices. These biases are subsequently amplified through matrix multiplication, leading to a cascading effect that ultimately undermines model robustness (Eq. 4). Therefore, *how to mitigate the negative effects of label noise during local LoRA fine-tuning* becomes a key challenge.

In FL, noisy updates from individual clients propagate through repeated aggregation, contaminating the global model and degrading system performance. Existing LoRA-based aggregation strategies, constrained by inflexibility or noise sensitivity, fail to suppress this amplification while maintaining model expressiveness. Therefore, *how to design a robust and flexible LoRA-based federated aggregation method to prevent noise amplification in federated communication* is an important issue.

In this paper, we propose RFedLR, a robust federated PEFT method under noisy scenarios, which consists of two stages: (1) To tackle the vulnerability of LoRA to label noise during local fine-tuning, we propose Sensitivity-aware Robust Tuning (SRT). SRT identifies and selectively updates noise-sensitive parameters during local training, thereby enhancing local model resilience to label noise while preserving stable features. (2) To overcome the aggregation discrepancies and noise amplification inherent in federated LoRA aggregation, we introduce Adaptive Federated LoRA Aggregation (AFLA). AFLA dynamically assesses the importance of local LoRA updates and assigns aggregation weights accordingly. This adaptive approach minimizes aggregation discrepancies and curtails noise propagation, leading to a more robust and accurate global model. The main contributions of this work are as follows:

- We first study and address the problem of robust federated PEFT in the presence of label noise.

- We propose RFedLR, a robust LoRA-based federated PEFT framework that leverages sensitivity-aware robust tuning and adaptive federated LoRA aggregation to mitigate the label noise effects.

- Extensive experiments demonstrate that RFedLR outperforms State-Of-The-Art (SOTA) methods, achieving higher accuracy and robustness in noisy federated learning scenarios.

## 2 Related Work

**Parameter-Efficient Fine-Tuning.** The computational and storage demands of FFT for large-scale models drive the development of PEFT methods, which update only a small subset of parameters or introduce limited trainable parameters. Adapter tuning [17] inserts small trainable adapter modules into each layer of a frozen pre-trained network. Prefix Tuning [18] and Prompt Tuning [19] append trainable continuous vectors to model inputs. BitFit [20] fine-tunes only the bias terms. Besides, LoRA [12], proposed by Hu *et al.*, is arguably the most widely adopted PEFT method to date. LoRA decomposes weight updates into low-rank matrices, significantly reducing trainable parameters without altering model architecture or input sequence length, thereby offering implementation simplicity and training stability. Furthermore, many variants of LoRA [21] are proposed. LoRA+ [22] employs an adjusted learning rate mechanism. DoRA [23] enhances tuning precision by decomposing weights into magnitude and direction components. LoRA-drop [24] and AdaLoRA [25] optimize adaptation through layer selection and dynamic rank allocation, respectively.

**Parameter-Efficient Fine-Tuning in FL.** Applying large-scale PTMs to FL is hindered by communication costs and local resource constraints. Consequently, several works explore the integration of PEFT methods within federated frameworks [26–28]. FS-LLM [29] provides a benchmark for federated fine-tuning of large language models. Zhang *et al.* [13, 30] evaluate prominent PEFT methods in FL, including Adapter Tuning (FedAP), Prefix Tuning (FedPF), BitFit (FedBF), and LoRA (FedLR). The results show that FedLR achieves competitive results while saving resource costs. SLoRA [14] tackles data heterogeneity via data-driven initialization and sparse tuning in a LoRA-based FL setting. Cho *et al.* [31] enable varying LoRA ranks across clients, with the server adaptively aggregating and distributing modules using zero padding and truncation. FlexLoRA [32] facilitates dynamic LoRA rank adjustments and utilizes Singular Value Decomposition (SVD) for post-aggregation weight assignment. These LoRA-based federated PEFT methods directly integrate LoRA into the standard FedAvg paradigm, which relies on weighted averaging for parameter aggregation. However, such direct LoRA aggregation can lead to interference. To address this, FFA-LoRA [15] updates only the zero-initialized matrix, while RoLoRA [16] alternates freezing low-rank matrices. Existing LoRA-based federated PEFT methods [33] exhibit limitations. They suffer from biases caused by direct LoRA aggregation or are limited by rigid update strategies, which can lead to suboptimal training results. Crucially, existing methods show significant performance degradation in the presence of label noise, which is a common problem in real-world FL deployments.

**Label Noise Learning.** Existing methods for label noise learning can be divided into four categories: (1) Noise transition matrix estimation. These approaches [34, 35] correct training by estimating a noise transition matrix. Dual T [36] decomposes the matrix to reduce estimation errors. Li *et al.* [37] exploit label correlation without anchors. (2) Sample selection and reweighting. Such techniques [38–41] identify reliable samples or adjust their weights during training. Han *et al.* [42, 43] train two networks to filter low-loss samples for cross-trains. DivideMix [44] leverages GMM and semi-supervised learning to separate noisy data. (3) Robust regularization. These methods [45] enhance model robustness via regularization schemes. Xia *et al.* [46] penalize non-critical parameters to prevent memorizing noisy labels. Menon *et al.* [47] propose a gradient clipping method based on composite loss to improve model robustness. (4) Robust loss functions. These functions [48–50] are designed for inherent noise tolerance. SCE [51] merges standard and reverse cross-entropy loss. Peer loss [52] uses predictions from a peer network to guide the current network.

**Label Noise Learning in FL.** Several works develop methods to mitigate label noise in FL. RHFL [53] studies the robust FL problem with label noise and heterogeneous model. Xu *et al.* [54] propose a multi-stage approach for label noise correction in FL. Wu *et al.* [55] focus on addressing class imbalance and label noise heterogeneity in FL. FedFixer [56] introduces a personalized model that cooperates with the global model to select clean samples. FedNed [57] uses the pseudo-label obtained by the global model to correct local training and performs aggregation through negative

distillation. While effective in conventional FL settings, these methods are not suitable for our federated PEFT environment. They usually assume access to sufficient local training resources or require massive data-level intervention, limiting their applicability in federated large-scale PTMs scenarios.

## 3 Motivation

### 3.1 Problem Setup and Notations

We consider a standard FL setting for a $C$-class image classification task involving $K$ clients. Each client $c_k$ possesses a private dataset $D_k = \{(x_i^k, y_i^k)\}_{i=1}^{N_k}$ with $N_k = | D_k |$, and $y_i^k \in \{0, 1\}^C$ is the one-hot ground-truth label for image $x_i^k$. However, in the noisy FL scenario we focus on, the labels can be incorrect. Thus, each client $c_k$ uses a noisy dataset $\tilde{D}_k = \{(x_i^k, \tilde{y}_i^k)\}_{i=1}^{N_k}$, where $\tilde{y}_i^k$ is the potentially corrupted label for $x_i^k$.

The federated fine-tuning process comprises iterations of local fine-tuning and collaborative aggregation, with $T$ denoting the communication rounds. The central server maintains a global pre-trained model with parameters $W_0$ and global fine-tuning parameters $\Delta W_0$. The pre-trained model parameters $W_0$ are pre-deployed on all client devices as $W_k$. During the FL process, only $\Delta W_0$ is updated, transferred, and aggregated, keeping the full parameter set $W_0$ frozen. In the local fine-tuning stage, clients receive the global fine-tuning parameters $\Delta W_0$. Each client $c_k$ fine-tunes the local model with the private dataset, obtains updated fine-tuning parameters $\Delta W_k$ and sends them to the server. In the collaborative aggregation phase, the server aggregates the fine-tuning parameter updates received from the clients. The process can be represented as

$$\mathcal{L}_k(\Delta W_k^t, \tilde{D}_k) = \frac{1}{N_k} \sum_{(x,\tilde{y}) \in \tilde{D}_k} \ell(W_0 + \Delta W_k^t; x, \tilde{y}),$$

$$\Delta W_0^{t+1} = \sum_{k=1}^{K} \frac{N_k}{N} \Delta W_k^t,$$

(1)

where $N = \sum_{k=1}^{K} N_k$ and $\ell(\cdot)$ is the loss function. Intuitively, our objective is to calculate an optimal set of local fine-tuning parameters $\Delta W = \{\Delta W_1, ..., \Delta W_K\}$ to minimize the total loss of all the clients. *The presence of noisy samples hinders local model convergence and destabilizes the global model upon aggregation due to error propagation. Therefore, it is crucial to investigate the noise-robust federated PEFT method.*

### 3.2 LoRA in Noisy Federated Learning

We adopt a widely-adopted PEFT method LoRA for federated fine-tuning, which exhibits superior performance compared to other PEFT methods [13]. LoRA decomposes the weight update matrix $\Delta W \in \mathbb{R}^{m \times n}$ in the pre-trained model into the product of two low-rank matrices $B \in \mathbb{R}^{m \times r}$ and $A \in \mathbb{R}^{r \times n}$, *i.e.*, $W_0 + \Delta W = W_0 + BA$, where the rank $r$ is significantly smaller than $m$ and $n$ ($r << min(m, n)$). This decomposition reduces the scale of trainable parameters from $O(m \times n)$ to $O(r \times (m + n))$. In practice, the modified forward pass through the layer is expressed as

$$W = W_0 + \frac{\alpha}{r} BA,$$

(2)

where $\alpha$ is a scaling hyperparameter. Typically, $A$ is randomly initialized, and $B$ is zero initialized, ensuring that $\Delta W$ is zero at the beginning of fine-tuning, thus preserving the initial performance of the pre-trained model.

However, directly integrating LoRA into FL introduces a fundamental aggregation challenge [15]. In the FL context, $W_0$ is pre-deployed to all clients and remains fixed, and only the LoRA matrices $A$ and $B$ are transmitted to the server for aggregation. Ideally, federated aggregation should operate on the full low-rank update, *i.e.* $\Delta W = BA$. The theoretically ideal aggregated update is shown in Eq. 3a. In contrast, the Vanilla FedAvg algorithm [2] separately averages the $A$ and $B$ matrices. The

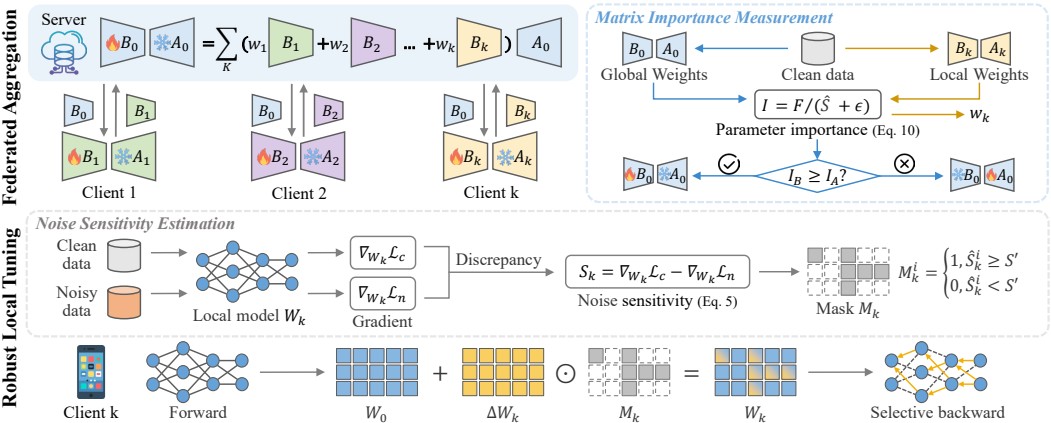

Figure 3: Illustration of RFedLR, which facilitates robust parameter-efficient fine-tuning for federated learning. In the local tuning phase, a sensitivity-aware robust tuning strategy is proposed to handle local noise amplification (§4.1). In the federated aggregation phase, We design the adaptive federated LoRA aggregation to prevent noisy information propagation in communication (§4.2).

actual aggregated update is shown in Eq. 3b as follows

$$\Delta W_0^+ = \frac{1}{K}(B_1 A_1 + B_2 A_2 + ... + B_K A_K), \tag{3a}$$

$$\neq \frac{1}{K}(B_1 + ... + B_K) \cdot \frac{1}{K}(A_1 + ... + A_K). \tag{3b}$$

Consequently, a discrepancy exists between the theoretically ideal aggregation (Eq. 3a) and the actual aggregation (Eq. 3b) that combine low-rank updates.

These challenges are compounded in noisy FL scenarios. The trainable parameter scale of LoRA is significantly less than FFT, which may render models more susceptible to overfitting noisy labels. More critically, label noise introduces bias terms into both low-rank matrices $A$ and $B$. The subsequent matrix multiplication non-linearly amplifies this bias as follows

$$\Delta W = (B + \delta B)(A + \delta A) = BA + B\delta A + \delta B A + \delta B \delta A, \tag{4}$$

where $\delta A$ and $\delta B$ denote the bias terms induced by noise. The noisy impact is not merely additive, but an additional noise term $\delta B \delta A$ is generated by matrix multiplication, which amplifies the noise effect. This degradation is further exacerbated in FL due to the iterative propagation of noisy information across clients. Our experiments (Fig. 2) demonstrate that label noise substantially impairs the performance of LoRA in FL, with the adverse impact intensifying over communication rounds. To address these compounded challenges, we propose RFedLR, which enhances robustness through sensitivity-aware local tuning and adaptive federated LoRA aggregation. The framework of RFedLR is described in Algorithm 1.

## 4 Proposed Method

### 4.1 Sensitivity-Aware Robust Tuning

To mitigate the adverse effects of label noise in federated LoRA fine-tuning, we propose SRT, which enables robust local fine-tuning through selective parameter updates based on noise sensitivity analysis (Fig. 3). The key insight behind SRT is that model parameters exhibit different degrees of sensitivity to label noise, with some parameters being extremely susceptible to noise-induced perturbations while others remain relatively stable. The parameters with low sensitivity are relatively stable and close to the ideal state, making them less susceptible to label noise. Conversely, highly sensitive parameters are more susceptible to noise and require careful fine-tuning. Therefore, we identify and selectively update parameters based on their noise sensitivity, improving the model robustness to noisy labels.

**Noise Sensitivity Estimation.** To quantify the sensitivity of individual parameters to label noise, we leverage contrastive analysis between clean and noisy learning dynamics to identify parameters that

exhibit high susceptibility to noise-induced perturbations. Specifically, the server collects a mini-batch clean public proxy dataset $D_c$, and a corresponding noisy dataset $D_n$ generated by injecting random label noise into $D_c$. The proxy datasets are sent to all clients. For each client $c_k$, we compute gradient vectors $\nabla_{W_k}\mathcal{L}_c(W_k, D_c)$ and $\nabla_{W_k}\mathcal{L}_n(W_k, D_n)$ of local model parameters $W_k$ on clean and noisy datasets, respectively. The magnitude of the gradient discrepancy between clean and noisy conditions for each parameter indexed by $i$ in the local model $W_k$ can be expressed as:

$$S_k^i = |\nabla_{W_k^i}\mathcal{L}_c - \nabla_{W_k^i}\mathcal{L}_n|. \tag{5}$$

The collection of all sensitivity scores forms a sensitivity tensor $S_k$, which has the same dimensions as its corresponding parameter tensor. The gradient magnitudes vary significantly across layers, and some layers that are sensitive to input changes may dominate the sensitivity estimation. To ensure comparability of gradient scales, we apply layer-wise min-max normalization to obtain the final sensitivity score. For parameter $i \in$ layer $l$, the final normalized sensitivity score $\hat{S}_k^i$ is

$$\hat{S}_k^i = \frac{S_k^i - min\{S_k^{(l)}\}}{max\{S_k^{(l)}\} - min\{S_k^{(l)}\}}. \tag{6}$$

where $S_k^{(l)}$ denotes the set of sensitivity scores for layer $l$. The parameters with high sensitivity scores show divergence in their optimization trajectories between noisy and clean data, indicating susceptibility to noise-induced biases. Conversely, a low score implies a more consistent update direction irrespective of label quality, signifying inherent stability.

**Sensitivity-Guided Gradient Propagation.** Based on the calculated parameter-level sensitivity scores, we selectively control gradient updates through a binary mask $M$. Given $\tau \in [0, 1]$ as the parameter keep ratio, it indicates the proportion of parameters allowed to be updated. For each parameter $W_k^i$, the corresponding mask element $M_k^i$ is determined as follows:

$$M_k^i = \begin{cases} 1, & \hat{S}_k^i \geq S' \\ 0, & \hat{S}_k^i < S' \end{cases} \tag{7}$$

where $S'$ is a threshold set to select the parameters with the top $\tau$ proportion of sensitivity scores. $\tau$ controls the proportion of parameters that allow updates, and $\tau = 1$ indicates standard LoRA fine-tuning. This mechanism differentiates between highly noise-sensitive and relatively stable parameters. We find that prioritizing updates for the most sensitive parameters significantly enhances model robustness. These highly sensitive parameters require recalibration to adapt to noisy environments, allowing the model to extract robust patterns. Meanwhile, preserving stable parameters helps the model maintain foundational representational capacity and mitigates overfitting to noisy samples, creating an implicit regularization effect. Importantly, during forward propagation, all parameters remain active to preserve model representability. Only in backward propagation, the mask $M$ affects the gradient flow by element-wise multiplication, similar to the conditional gradient dropout [58]. In round $t \in [0, T]$, the local fine-tuning process on the private noisy dataset $\tilde{D}_k$ can be formulated as:

$$\Delta W_k^{t+1} = \Delta W_k^t - \eta \nabla \mathcal{L}_{ce}(\Delta W_k^t, \tilde{D}_k) \odot M_k^t, \tag{8}$$

where $\eta$ is the learning rate, $\mathcal{L}_{ce}$ denotes the cross-entropy loss and $\odot$ denotes element-wise multiplication. This selective updating strategy improves the noise robustness of local fine-tuning and reduces the computational cost by limiting the amount of updated parameters.

## 4.2 Adaptive Federated LoRA Aggregation

While SRT improves the robustness of local fine-tuning, effectively aggregating LoRA updates under label noise remains challenging. Existing static or alternating matrix update schemes fail to capture dynamic parameter importance and stability. Our proposed AFLA addresses this by dynamically updating and aggregating LoRA matrices based on these properties (Fig. 3).

**Matrix Importance Measurement.** To discern the significance of matrices $A$ and $B$ within the local update $\Delta W_k = B_k A_k$ for each client $c_k$, we propose a matrix importance measurement that considers both parameter information and noise sensitivity. We adopt the Fisher Information Matrix (FIM) to calculate parameter contribution, which effectively estimates the information content of parameters [59, 60]. The parameters with higher Fisher values are considered more informative,

as they contribute more significantly to the model decisions. However, local noise may distort the gradient calculation, resulting in a biased Fisher information estimate. To accurately evaluate the parameter contribution, we calculate the Fisher information value on the clean public proxy dataset $D_c$ instead of the local noisy dataset $\tilde{D}_k$. Since computing the exact FIM is computationally expensive, we adopt an empirical Fisher approximation. Specifically, the FIM measures the parameter contribution to the curvature of the loss function by calculating the square of the first-order derivative of the log-likelihood function. The Fisher information for an individual parameter indexed by $i$ in the local model $W_k$ is defined as:

$$F_k^i = \left(\frac{\partial \log \mathcal{L}(W_k, D_c)}{\partial W_k^i}\right)^2. \tag{9}$$

We then integrate both noise sensitivity and parameter contribution into a hybrid importance score $I$. The matrix with lower noise sensitivity (Eq. 5) is more stable and reliable for federated aggregation. The hybrid importance score can be expressed as:

$$I_k^i = \frac{F_k^i}{S_k^i + \epsilon}, \tag{10}$$

where $\epsilon = 1e - 8$ ensures computational stability. To ensure the comparability of the importance score across layers, we apply the layer-wise min-max normalization to $I_k^i$ to obtain $\hat{I}_k^i$. Without normalization, the inherent input-sensitive layer parameters will always have higher importance.

The overall importance scores for the update matrices $B$ and $A$ of client $c_k$, denoted as $I_B$ and $I_A$ respectively, are obtained by averaging the normalized parameter importance scores within each individual matrix:

$$I_{A_k} = \frac{1}{|A_k|} \sum_{i \in A_k} \hat{I}_{A_k}^i, I_{B_k} = \frac{1}{|B_k|} \sum_{j \in B_k} \hat{I}_{B_k}^j. \tag{11}$$

**Selective Weighted Aggregation.** To mitigate aggregation bias and noise amplification in federated aggregation, we introduce a selective weighted aggregation strategy. This approach dynamically identifies the trainable matrix based on the matrix importance of the global model. Before each FL round $t$, we calculate the matrix importance $I_{A_0^t}$ and $I_{B_0^t}$ for the global model to determine the update matrix. The trainable matrix $X^t$ is selected as follows:

$$X^t = \begin{cases} A, & \text{if } I_{A_0^t} > I_{B_0^t} \\ B, & \text{otherwise} \end{cases}, \tag{12}$$

and the other LoRA matrix, denoted as $Y^t$. The adaptive selection prioritizes the high-importance matrix while keeping the other matrix frozen. During the local fine-tuning phase, clients only fine-tune the selected matrix $X^t$. This strategy concentrates computational resources on the most impactful parameter updates, minimizing aggregation bias and reducing noise propagation.

After the clients perform robust local fine-tuning (Sec. 4.1), we recalculate the updated trainable matrix importance score for clients to guide aggregation. In the federated aggregation phase, clients upload $X_k^t$ to the server, and the server then aggregates these received low rank matrices via a weighted average scheme. This scheme comprehensively considers both the trainable matrix importance and the local data scale of each client. Specifically, the federated aggregation process for the round $t \in [0, T]$ can be formulated as:

$$X_0^{t+1} = \sum_k w_k^t X_k^t,$$

$$w_k^t = \lambda \frac{I_{X_k^t}}{\sum_{i=1}^K I_{X_i^t}} + (1 - \lambda) \frac{N_k}{\sum_{i=1}^K N_i}, \tag{13}$$

where $w_k$ is the weight for client $c_k$ and $\lambda$ balances client importance and data scale. $I_{X_k}$ denotes the importance score of the selected trainable update matrix of client $c_k$ in current round. By integrating data scale and parameter importance into the weighting scheme, we achieve efficient and robust knowledge transfer. By implicitly reducing the weight of contributions from potentially noisy or less informative updates, this approach effectively mitigates the detrimental impact of label noise in FL. Following aggregation, the server transmits only this single, updated global low-rank matrix $X_0^{t+1}$ back to all participating clients. This download step only transmits a single matrix, thus halving the communication overhead.

# 5  Experiments

## 5.1  Experimental Setup

**Datasets and Models.** Following previous works [53, 61], we conduct extensive experiments on CIFAR-100 [62] dataset, which contains $60,000$ color images covering 100 classes. The FL setup involved $K = 5$ clients, each holding a non-IID partition of CIFAR-100. The private data is partitioned among clients according to a Dirichlet distribution, with the degree of data heterogeneity controlled by the hyperparameter $\beta$, which is set to $0.5$. We maintain a mini-batch of clean CIFAR-100 dataset on the server as the public proxy dataset $D_c$ with $|D_c| = 256$. Following existing visual PEFT works [63], we set an initial global model on the server, which adopts Vision Transformer-Base (ViT-B) [64] as the backbone architecture, initialized with ImageNet-21K [65] pre-trained weights, processing $224 \times 224$ pixel images with a $16 \times 16$ patch size.

**Label Noise Settings.** We investigate two common settings for label noise [42]. (1) pair flip (pairflip) noise, flipping the label to a pre-defined similar class. (2) symmetric flip (symflip) noise, randomly flipping the label to any other class with uniform probability. We experiment with noise rate $\mu$ of $0.2$ and $0.4$. Clients sample randomly from shuffled noisy CIFAR-100, so the noise rates of different clients might be inconsistent to simulate practical settings.

**Comparison Methods.** We compare RFedLR with SOTA methods to demonstrate its effectiveness in noisy FL setting. FedPF [13] applies prefix tuning in federated settings. FedBF [13] only trains and share bias parameters. FedAP [13] updates only small adapter modules inserted between model layers. FedLR [13] incorporates LoRA into FL. SLoRA [14] performs data-driven initialization and sparse fine-tuning to handle data heterogeneity in FL. FFA-LoRA [15] only updates and aggregates the zero-initialized matrix in LoRA. RoLoRA [16] alternately freezes different low-rank matrices during communication rounds. FlexLoRA [32] implements weight redistribution through SVD. All methods were implemented within a unified PyTorch framework for fair comparison.

**Implementation Details.** The FL system comprises $K = 5$ clients and runs for $T = 40$ communication rounds. Local fine-tuning is performed for one epoch per communication round. For optimization, we use SGD with a learning rate of $0.01$, weight decay of $0.0001$, momentum of $0.9$ and a batch size of $256$. LoRA decomposes weight updates into two low-rank matrices $A$ and $B$ with rank $r = 4$, and the scaling factor $\alpha$ is set to $4$. SRT employed a keep ratio $\tau = 0.2$. For AFLA, the balancing hyperparameter $\lambda$ is set to $0.4$. Experiments are conducted on 4 NVIDIA RTX 3090 GPUs.

## 5.2  Ablation Study

As shown in Tab. 1, we evaluate the individual and combined effectiveness of all components under noise ratios of 0.2, 0.4, and noise types of pairflip and symflip. The baseline (first row in Tab. 1) represents Vanilla FedAvg combined with LoRA.

**Effectiveness of SRT.** SRT mitigates local noise by selectively updating parameters based on their estimated noise sensitivity. The results in Tab. 1 show that enabling SRT alone yields notable improvements over the baseline across all noise conditions. For $\mu = 0.2$, the average accuracy increases from 75.28% to 81.63% (pairflip) and from 83.65% to 85.03% (symflip). At $\mu = 0.4$, SRT achieves 64.13% for pairflip noise and 84.62% for symflip noise, yielding improvements of 7.51% and 4.93%, respectively. The efficacy of SRT is pronounced at higher noise rates, affirming its effectiveness in countering local noise-induced perturbations. Besides, the impact of the hyperparameter $\tau$ on SRT is further investigated through a sensitivity analysis detailed in Appendix B.1.

**Effectiveness of AFLA.** AFLA adaptively freezes a single matrix and dynamically weights clients, aiming to minimize bias and noise amplification during global aggregation. We evaluate its contribution by comparing the baseline model against the model with only AFLA enabled. For $\mu = 0.2$, AFLA improves the baseline accuracy from 75.28% to 76.07% under pairflip noise and from 83.65% to 84.87% under symflip noise. At $\mu = 0.4$, AFLA also shows significant improvements. The results demonstrate the ability of AFLA to refine aggregation by prioritizing stable and informative updates. As shown in Appendix B.2, we implemented ablation experiments to analyze the respective contributions of the Fisher information component and the noise sensitivity component of the hybrid

Table 1: Ablation study §5.2 of each component. The average test accuracy (%) of local models is demonstrated.

| Components | | $\mu = 0.2$ | | $\mu = 0.4$ | |
|---|---|---|---|---|---|
| SRT | AFLA | Pairflip | Symflip | Pairflip | Symflip |
| | | 75.28 | 83.65 | 56.62 | 79.69 |
| ✓ | | 81.63 | 85.03 | 64.13 | 84.62 |
| | ✓ | 76.07 | 84.87 | 57.19 | 81.36 |
| ✓ | ✓ | **83.12** | **86.97** | **67.08** | **84.64** |

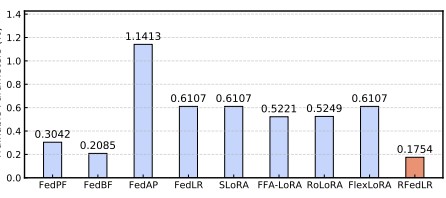

Figure 4: Comparison of trainable parameter (%) for various methods relative to FFT.

importance score in AFLA. Furthermore, we conduct a sensitivity analysis on $\lambda$ in Appendix B.1 to demonstrate the effectiveness of AFLA.

**Combined Effectiveness of SRT and AFLA.** The RFedLR framework integrates SRT and AFLA to achieve the highest performance in all settings. Especially at $\mu = 0.4$, accuracy under pairflip noise and symflip noise improves by 10.46% and 4.95% from baseline. The performance of the full model surpasses the individual contributions of SRT and AFLA, revealing a synergistic effect.

## 5.3 Comparison with SOTA Methods

We evaluate the performance of RFedLR against SOTA federated PEFT methods. As shown in Tabs. 2 and 3, model performance is evaluated through the test accuracy of local models, across different noise rates. LoRA-based methods consistently outperform other PEFT approaches, especially under higher noise rates. This reinforces the suitability of LoRA as a foundation for federated fine-tuning in noisy environments. RFedLR builds upon this foundation, leveraging SRT and AFLA to achieve further improvements. RFedLR consistently outperformed all SOTA methods, demonstrating strong robustness in noisy FL settings. The superior performance of RFedLR across both noise types and rates can be attributed to SRT and AFLA. SRT enhances local tuning by selectively updating parameters based on noise sensitivity, reducing overfitting to noisy labels. Meanwhile, AFLA mitigates aggregation bias and noise propagation through adaptive matrix aggregation and dynamic weighting based on importance and stability, ensuring efficient and robust global updates. The larger performance gains under pairflip noise, especially at $\mu = 0.4$, where RFedLR outperforms RoLoRA by 10.15%, suggest that RFedLR excels in handling structured noise and the performance advantage is more pronounced at increased noise intensity.

All methods performed worse under pairflip noise than symflip, due to differing disruption mechanisms. Symflip noise introduces uniform, less correlated errors, while pairflip creates structured, harder-to-resolve noise by flipping labels to similar classes. Nonetheless, RFedLR achieves significant improvements under both pairflip and symflip noise. Under pairflip noise with $\mu = 0.4$, RFedLR achieves an average accuracy of 67.08%, surpassing the existing best method RoLoRA by 10.15%. Under symflip noise with $\mu = 0.4$, RFedLR achieves 84.64%, which is 3.13% higher than SLoRA.

In terms of parameter efficiency, RFedLR also demonstrates significant advantages, which is crucial for deployment on resource-constrained edge devices. As illustrated in Fig. 4, RFedLR utilizes the fewest trainable parameters (0.1754% relative to Full Fine-Tuning) among all compared methods. This efficiency makes RFedLR well-suited for resource-constrained FL scenarios.

Beyond the main vision tasks, we further validate the generalizability and robustness of our framework in additional experiments detailed in Appendix B.3. To assess its versatility, we test RFedLR on the MNLI Natural Language Processing (NLP) task, where it outperforms other federated PEFT methods. Besides, we validate the scalability of RFedLR in larger-scale cross-device scenarios (see Appendix B.4 for details). Furthermore, we compare our approach against SOTA federated noise learning methods, by adapting their core noise-handling mechanisms to the PEFT setting for a fair comparison. As shown in Appendix B.5, RFedLR achieves competitive performance.

## 5.4 Analysis of the Proxy Dataset

Our method preserves the foundational privacy guarantee of federated learning, *i.e.*, no private client data is ever shared. We follow the published federated learning research [66] in setting up proxy datasets, which can come from small public datasets relevant to the task, or from trusted participants in the federated learning system. In our work, we require only a mini-batch dataset to serve this

Table 2: Comparison with the SOTA methods §5.3 when the noise rate $\mu = 0.2$.

| Method | Pairflip | | | | | | Symflip | | | | | |
|---|---|---|---|---|---|---|---|---|---|---|---|---|
| | $W_1$ | $W_2$ | $W_3$ | $W_4$ | $W_5$ | Avg | $W_1$ | $W_2$ | $W_3$ | $W_4$ | $W_5$ | Avg |
| FedPF [13] | 68.29 | 67.95 | 67.45 | 68.87 | 67.32 | 67.98 | 75.45 | 76.10 | 74.25 | 76.47 | 75.94 | 75.64 |
| FedBF [13] | 75.87 | 74.74 | 73.47 | 75.26 | 74.56 | 74.78 | 83.74 | 82.68 | 82.35 | 83.67 | 84.17 | 83.32 |
| FedAP [13] | 75.58 | 75.79 | 73.83 | 76.93 | 75.28 | 75.48 | 83.02 | 82.20 | 81.46 | 83.01 | 84.38 | 82.81 |
| FedLR [13] | 77.09 | 73.85 | 73.50 | 75.98 | 75.97 | 75.28 | 84.50 | 83.03 | 82.87 | 83.94 | 83.90 | 83.65 |
| SLoRA [14] | 77.12 | 76.18 | 76.17 | 77.16 | 75.78 | 76.48 | 84.89 | 83.97 | 84.07 | 85.06 | 84.51 | 84.50 |
| FFA-LoRA [15] | 73.68 | 72.68 | 72.05 | 74.65 | 72.94 | 73.20 | 81.09 | 80.59 | 79.05 | 80.94 | 80.34 | 80.40 |
| RoLoRA [16] | 76.62 | 74.88 | 74.42 | 76.48 | 76.22 | 75.73 | 85.21 | 83.95 | 83.69 | 84.86 | 85.03 | 84.55 |
| FlexLoRA [32] | 75.94 | 73.68 | 71.77 | 76.00 | 76.01 | 74.68 | 84.39 | 83.29 | 82.47 | 83.75 | 84.16 | 83.61 |
| **RFedLR** | **83.40** | **82.96** | **83.28** | **82.88** | **83.07** | **83.12** | **87.17** | **86.69** | **86.99** | **87.00** | **87.02** | **86.97** |

Table 3: Comparison with the SOTA methods §5.3 when the noise rate $\mu = 0.4$.

| Method | Pairflip | | | | | | Symflip | | | | | |
|---|---|---|---|---|---|---|---|---|---|---|---|---|
| | $W_1$ | $W_2$ | $W_3$ | $W_4$ | $W_5$ | Avg | $W_1$ | $W_2$ | $W_3$ | $W_4$ | $W_5$ | Avg |
| FedPF [13] | 50.27 | 50.26 | 48.80 | 51.19 | 50.00 | 50.10 | 72.30 | 73.66 | 72.45 | 72.778 | 72.76 | 72.79 |
| FedBF [13] | 55.76 | 55.00 | 53.46 | 56.24 | 59.30 | 55.95 | 79.11 | 77.69 | 76.98 | 78.30 | 80.43 | 78.50 |
| FedAP [13] | 56.56 | 54.20 | 52.81 | 57.03 | 57.40 | 55.60 | 78.98 | 77.89 | 77.15 | 78.69 | 80.40 | 78.62 |
| FedLR [13] | 56.91 | 55.91 | 56.88 | 57.56 | 55.83 | 56.62 | 79.31 | 79.33 | 79.13 | 79.22 | 81.46 | 79.69 |
| SLoRA [14] | 54.30 | 56.38 | 54.58 | 58.06 | 60.55 | 56.77 | 81.36 | 81.38 | 80.55 | 81.78 | 82.45 | 81.51 |
| FFA-LoRA [15] | 54.81 | 54.90 | 54.33 | 56.62 | 53.22 | 54.78 | 77.05 | 76.61 | 77.53 | 78.01 | 77.87 | 77.41 |
| RoLoRA [16] | 57.18 | 55.75 | 57.39 | 58.20 | 56.13 | 56.93 | 80.95 | 81.00 | 80.79 | 81.53 | 82.27 | 81.31 |
| FlexLoRA [32] | 56.56 | 55.46 | 56.24 | 58.04 | 56.60 | 56.58 | 79.58 | 78.97 | 78.83 | 79.66 | 81.49 | 79.71 |
| **RFedLR** | **67.36** | **67.04** | **66.82** | **67.04** | **67.13** | **67.08** | **83.83** | **84.61** | **84.41** | **85.69** | **84.67** | **84.64** |

purpose. Crucially, the effectiveness of our method does not depend on the assumption that the distribution or noise pattern of the proxy dataset must match to the local data.

**Robustness to Data Distribution Mismatch.** In our setup, the proxy dataset is a uniformly distributed mini-batch randomly sampled from the CIFAR-100 test set. However, the local data was explicitly configured to be non-IID. As demonstrated by our results in Tabs. 2 and 3, particularly in the highly data heterogeneous environments (Appendix B.6), our method performs well, which indicates that distribution matching is unnecessary.

**Robustness to Noise Type Mismatch.** The noisy proxy dataset $D_n$ is generated by injecting symflip noise into the clean dataset. Symflip noise represents a general form of label noise that does not rely on any assumptions about class similarity. This ensures that our sensitivity analysis is robust and not biased towards a specific structured noise pattern. Tabs. 2 and 3 show that RFedLR still demonstrates remarkable effectiveness when clients have structured pairflip noise. Besides, Tab. 1 shows that the introduction of SRT module results in a significant accuracy improvement under pairflip noise with different noise rates. This shows that a mismatch between the proxy noise and the local true noise does not undermine the effectiveness of our gradient discrepancy analysis.

The proxy dataset serves as an objective benchmark for identifying parameters that are unstable to noise. This instability is a general property of model parameters, rather than their response to specific data distributions or noise patterns. Furthermore, the results in Appendix B.7 show that even using task-irrelevant proxy datasets is still sufficient to detect the most significant sensitive parameters.

# 6 Conclusion

In this paper, we tackle the challenges of fine-tuning large-scale PTMs in FL under label noise scenarios. We propose RFedLR, a robust PEFT framework including SRT and AFLA. SRT enhances local model robustness by selectively updating noise-sensitive parameters. AFLA mitigates aggregation discrepancies and curbs noise propagation via dynamic LoRA update aggregation. Comprehensive experiments conducted in noisy FL settings demonstrate that RFedLR achieves superior accuracy and robustness compared to SOTA methods, advancing the deployment of large-scale models in such environments. The primary limitations of RFedLR include its reliance on a mini-batch public proxy dataset for estimating noise sensitivity and matrix importance, a dependency that may constrain broader applicability. Furthermore, while RFedLR achieves substantial reductions in trainable parameters, the requisite computations for noise sensitivity estimation and importance measurement introduce additional resource demands. Future work will focus on developing proxy-data-free and more computationally efficient solutions.

**Acknowledgements.** This work is partially supported by the National Key Research and Development Program of China (2024YFC3308400), National Natural Science Foundation of China under Grant (62361166629, 62176188), Major Project of Science and Technology Innovation of Hubei Province (2024BCA003, 2025BEA002).

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

# A  The Algorithm of RFedLR

We summarize the Pseudocode of RFedLR in Algorithm 1.

---

**Algorithm 1:** Training process of RFedLR

---

**Input:** Total communication rounds $T$, number of clients $K$, private noisy dataset $\tilde{D}_k$ for each
client $k$, global public clean proxy dataset $D_c$ and its noisy version $D_n$, global
pre-trained model parameters $W_0$, initial global LoRA matrices $A_0^0, B_0^0$, learning rate $\eta$.

**Server Execute:**
**for** *each communication round $t = 0, 1, \ldots, T-1$* **do**

    Calculate importance scores $I_{A_0^t}, I_{B_0^t}$ by Eqs. (10) and (11)

    Select trainable matrix $X^t$ with higher importance and the other is frozen matrix $Y^t$

    **for** *each client $k = 1, 2, \ldots, K$ in parallel* **do**

        Send $A_0^t, B_0^t, D_c, D_n$, and the selection $X^t$ to client $k$

        $X_k^{t+1}, I_{X_k^{t+1}} \leftarrow LocalUpdate(A_0^t, B_0^t, D_c, D_n, X^t)$

    **end**

    Calculate aggregation weight $w_k^t$ and aggregate the received matrices $X_0^{t+1}$ by Eq. (13)

**end**

**Local Update:**
**Function** *LocalUpdate($A_0^t, B_0^t, D_c, D_n, X^t$)*

    Receive $A_0^t, B_0^t, D_c, D_n$, and selection $X^t$ from Server

    Initialize local matrices $A_k^t \leftarrow A_0^t, B_k^t \leftarrow B_0^t$. Let local model be $W_k^t = W_0 + B_k^t A_k^t$

    Compute gradients on proxy data: $\nabla_{W_k} \mathcal{L}_c(W_k^t, D_c)$ and $\nabla_{W_k} \mathcal{L}_n(W_k^t, D_n)$

    Calculate the normalized parameter sensitivity score $\hat{S}_k$ by Eqs. (5) and (6)

    Generate a binary mask $M_k^t$ by Eq. (7)

    Update only the trainable matrix $X_k^t$ by Eq. (8)

    Recalculate the importance score $I_{X_k^{t+1}}$ for the updated matrix $X_k^{t+1}$

    **return** $X_k^{t+1}, I_{X_k^{t+1}}$

---

# B  Additional Experimental Results

## B.1  Hyperparameter Sensitivity Analysis

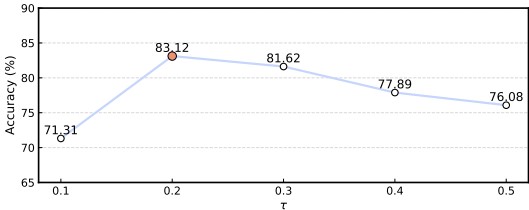

Figure 5: Parameter analysis for $\tau$ in the SRT module. The analysis is conducted under the 20% pairflip noise
setting.

We include a sensitivity analysis for the key hyperparameters $\tau$ and $\lambda$. For the parameter keep rate $\tau$
in SRT (Fig. 5), our analysis shows that performance peaks at our chosen value of $\tau = 0.2$, achieving
an accuracy of $83.12\%$. When $\tau$ is too low, the model plasticity is limited. When $\tau$ is too high, the
regularization effect of the selective update strategy is weakened and performance decline.

For the balancing hyperparameter $\lambda$ in AFLA (Tab. 4), our results show that the performance of
RFedLR is relatively stable for the selection of $\lambda$. When $\lambda$ is 0.4, RFedLR achieves the highest test
accuracy, which can effectively balance the contribution of data scale and matrix importance.

Table 4: Parameter analysis for $\lambda$ in the AFLA module. The analysis is conducted under the 20% symflip noise setting.

| $\lambda$ | 0.1 | 0.2 | 0.3 | **0.4** | 0.5 | 0.6 | 0.7 | 0.8 |
|---|---|---|---|---|---|---|---|---|
| $Acc$ | 86.22 | 85.50 | 85.46 | **86.97** | 85.34 | 85.84 | 85.42 | 86.06 |

## B.2 Component Analysis of the Hybrid Importance Score in AFLA

Table 5: Ablation study on the components of the hybrid importance score. The table compares the average test accuracy (%) of RFedLR (Ours) with variants that exclude Fisher information (w/o F) or noise sensitivity (w/o S) from the importance calculation in Eq. (10).

| Method | $\mu = 0.2$ | | $\mu = 0.4$ | |
|---|---|---|---|---|
| | Pairflip | Symflip | Pairflip | Symflip |
| **Ours** | **83.12** | **86.97** | **67.08** | **84.64** |
| w/o F | 82.75 | 85.23 | 66.69 | 84.38 |
| w/o S | 77.42 | 80.86 | 61.04 | 77.44 |

The hybrid importance score combines FIM with our noise sensitivity metric. We adopt the empirical FIM to quantify the parameter importance, as it is a classic and effective measure of parameter impact on model prediction. Parameters with high Fisher Information values most affect predictive ability. Although FIM can identify important parameters under ideal conditions, it ignores the parameter stability in noisy environments. Therefore, we design the hybrid importance score $I = F/(S + \epsilon)$ to balance the contribution and reliability of parameters.

To validate this design choice, we supplement this with ablation experiments (Tab. 5) to analyze the respective contributions of F and S to the final performance. We compare the RFedLR method ("Ours") with two variants: one that removes the Fisher information component ("w/o F," using $I = 1/(S + \epsilon)$) and the other that removes the noise sensitivity component ("w/o S," using $I = F$).

The results confirm that both components contribute positively, justifying our combined formulation. The significant performance drop in the "w/o S" case confirms that accounting for noise sensitivity is crucial for robust aggregation. Using Fisher information alone is insufficient and can be misleading in noisy environments. The full method always outperforms the "w/o F" case, which indicates that Fisher information further refines the importance score. The superiority of this hybrid score over using either Fisher information or noise sensitivity alone.

## B.3 Effectiveness on NLP Tasks

Table 6: Comparison with SOTA methods on MNLI task from GLUE benchmark when the noise rate is 0.2 and noise type is symflip. We take the average test accuracy (%) of local models for demonstration.

| Method | FedPF | FedBF | FedAP | FedLR | SLoRA | FFA-LoRA | RoLoRA | FlexLoRA | **RFedLR** |
|---|---|---|---|---|---|---|---|---|---|
| $Acc$ | 66.45 | 67.64 | 67.04 | 67.60 | 69.46 | 64.90 | 68.20 | 67.60 | **71.42** |

To validate the generalizability of RFedLR, we conduct experiments on NLP tasks. Following existing federated PEFT works [13, 15, 67], we perform an experiment on the MNLI [68] task from the GLUE [69] benchmark. We utilize Roberta-Base [70] as the initial global model. The experimental setup mirrored our vision experiments, with clients holding non-IID data partitions and operating under a 20% symflip label noise. As shown in the Tab. 6, RFedLR achieves a accuracy of 71.42%, significantly outperforming all baseline methods. The results confirm that our framework is effective for NLP tasks in noisy federated settings.

## B.4 Performance in Large-Scale Scenarios

Table 7: Performance Comparison in a large-scale federated setting (100 total clients, 10% participation rate) with 40% pairflip noise. We take the average test accuracy of the local models for demonstration.

| Method | FedLR | SLoRA | FFA-LoRA | RoLoRA | FlexLoRA | **RFedLR** |
|---|---|---|---|---|---|---|
| $Acc$ | 59.04 | 53.91 | 58.74 | 58.75 | 54.17 | **59.61** |

Validating the scalability and generality of our method in a setting with a larger number of clients is essential for demonstrating its practical applicability. Therefore, we conduct experiments to simulate a more realistic, large-scale cross-device scenario. This experiment involves 100 total clients with a 10% participation rate, meaning 10 clients are sampled for training and aggregation in each

communication round. The results in Tab. 7 show that RFedLR framework maintains its performance advantage in large-scale systems.

## B.5 Comparison with SOTA Federated Noise Learning Methods

Table 8: Performance comparison of RFedLR with SOTA robust federated learning methods under a noise rate of 0.4. Values represent average test accuracy (%).

| NoiseType | FedCorr | RHFL | FedFixer | **RFedLR** |
|---|---|---|---|---|
| Pairflip | 55.82 | 57.44 | 55.61 | **67.08** |
| Symflip | 79.80 | **84.72** | 77.82 | 84.64 |

The experiments in Tabs. 2 and 3 focused on comparing RFedLR with other federated PEFT methods. Additionally, We implement and evaluate robust federated learning methods, FedCorr [54], RHFL [53], FedFixer [56]. As these methods are for conventional full-parameter FL, we adapt their core noise-handling mechanisms to our federated PEFT setting for a fair comparison. The experiments are performed under a noise rate of 0.4. These results (Tab. 8) demonstrate that RFedLR outperforms these robust FL methods. We analyze that these methods cannot mitigate the aggregation bias inherent in federated LoRA, while our framework is designed with SRT to counter local noise amplification and AFLA to address aggregation bias and noise propagation.

## B.6 Performance under Severe Data Heterogeneity

Table 9: Comparison with SOTA methods under severe data heterogeneity (Dirichlet concentration parameter is 0.1) when the noise rate is 0.4. We take the average test accuracy (%) of local models for demonstration.

| NoiseType | FedPF | FedBF | FedAP | FedLR | SLoRA | FFA-LoRA | RoLoRA | FlexLoRA | **RFedLR** |
|---|---|---|---|---|---|---|---|---|---|
| Pairflip | 42.62 | 50.61 | 50.98 | 51.90 | 50.83 | 48.38 | 52.06 | 51.75 | **62.56** |
| Symflip | 63.22 | 71.70 | 62.56 | 73.01 | 75.24 | 68.55 | 74.72 | 72.33 | **80.48** |

We have conduct the experiments in a highly non-IID setting by setting the Dirichlet concentration parameter to 0.1, with a noise rate of 0.4. The results (Tab. 9) show that the performance advantage of RFedLR becomes even more pronounced in this severe data heterogeneous scenario. We analyze the advantages to be attributed to the AFLA mechanism. AFLA weights clients based on the amount of data, and the estimated importance and stability of their LoRA updates, ensuring robust aggregation results in highly non-IID scenarios.

## B.7 Robustness to Mismatched Proxy Datasets

Table 10: Performance comparison with the SOTA methods, where the local dataset is set to a subset of CIFAR-100. "RFedLR(CIFAR-10)" and "RFedLR(CIFAR-100)" respectively indicate that we use a subset of CIFAR-10 and CIFAR-100 as the proxy dataset. The best and second-best results are highlighted in bold.

| Method | $\mu = 0.2$ | | $\mu = 0.4$ | |
|---|---|---|---|---|
| | Pairflip | Symflip | Pairflip | Symflip |
| FedPF | 67.98 | 75.64 | 50.10 | 72.79 |
| FedBF | 74.78 | 83.32 | 55.95 | 78.50 |
| FedAP | 75.48 | 82.81 | 55.60 | 78.62 |
| FedLR | 75.28 | 83.65 | 56.62 | 79.69 |
| SLoRA | 76.48 | 84.50 | 56.77 | 81.51 |
| FFA-LoRA | 73.20 | 80.40 | 54.78 | 77.41 |
| RoLoRA | 75.73 | **84.55** | 56.93 | 81.31 |
| FlexLoRA | 74.68 | 83.61 | 56.58 | 79.71 |
| **RFedLR (CIFAR-10)** | **76.70** | 84.54 | **57.22** | **81.96** |
| **RFedLR (CIFAR-100)** | 83.12 | 86.97 | 67.08 | 84.64 |

To validate the generalizability and adaptability of our proposed method, we supplement experiments by setting the proxy dataset to CIFAR-10, while the local dataset remain CIFAR-100. This setup introduces a significant mismatch in the label space and task complexity. The results in Tab. 10 show that the performance of RFedLR is reduced when using the mismatched proxy (CIFAR-10) compared to using the matched proxy (CIFAR-100). However, even with this significant mismatch, RFedLR remains robust. RFedLR (CIFAR-10) still outperforms other state-of-the-art methods in various scenarios. The purpose of using the proxy dataset is to estimate the parameter sensitivity to gradient perturbations induced by label noise. While a task-irrelevant proxy dataset may not perfectly capture parameter sensitivity, it is still sufficient to detect the most significant instabilities, thereby identifying the parameters most susceptible to noise-induced instability.

