# OpenReview forum: "Towards Robust Parameter-Efficient Fine-Tuning for Federated Learning"
_NeurIPS.cc/2025/Conference — NeurIPS 2025 poster_

### Official Review · Reviewer_Egmb · 2025-06-27

**Clarity:** 2
**Significance:** 2
**Originality:** 3
**Rating:** 4
**Confidence:** 3

**Summary:**

This paper proposes RFedLR, a robust parameter-efficient fine-tuning method for federated learning (FL) under label noise.

The method has two main components:
- (1) Sensitivity-aware robust tuning (SRT), where each client estimates parameter-wise noise sensitivity using gradients from a clean proxy dataset and masks updates to the most noise-sensitive parameters; and
- (2) Adaptive Federated LoRA Aggregation (AFLA), which aggregates LoRA parameters from clients using weights based on both client data size and empirical importance scores derived from the Fisher information.

Experiments on CIFAR-100 with synthetic label noise show that RFedLR consistently outperforms several PEFT and LoRA-based FL baselines, while maintaining a low communication and training footprint.

**Questions:**

1. Could the proposed framework be extended to NLP tasks, such as LLM fine-tuning? What are the main challenges in applying it beyond image classification?

2. The notations in Equations (5)–(7) are unclear—are these quantities matrices, vectors, or scalars? Including dimension annotations or explanations in the text would improve readability.

3. The aggregation weights in AFLA (Eq. 13) appear intuitive but lack theoretical support. Could the authors provide either a toy example or empirical evidence to show how the weighting scheme helps mitigate noise amplification? Even a high-level, non-rigorous explanation would be helpful.

4. Figure 2’s caption should clarify what is meant by "federated LoRA." Based on the results, its performance under moderate noise does not appear significantly degraded, which seems to contradict statements made in the main text.

5. Could the authors provide more justification for the design of the overall importance score in Eq. (10)? The formulation feels heuristic—some explanation or ablation would be useful to understand its effect.

**Ethical Concerns:**

["NO or VERY MINOR ethics concerns only"]

**Final Justification:**

All of my questions and concerns have been addressed. I would increase my points.

**Limitations:**

The authors mention reliance on a small clean proxy set and acknowledge computational overhead for sensitivity estimation.

It would be helpful to further discuss limitations such as scalability to many clients, applicability to other tasks (e.g., NLP), and the potential privacy implications of using proxy gradients.

**Paper Formatting Concerns:**

N/A.

**Quality:**

2

**Strengths And Weaknesses:**

**Strengths**

1. The paper addresses an important challenge in FL where label noise can severely degrade model performance, especially under parameter-efficient fine-tuning.

2. The method is practical, requiring only a small clean proxy set, and shows strong empirical performance with well-designed ablations.

3. The paper is generally clear and well-structured, though some notation is hard to follow.

**Weaknesses**

1. The connection to LLM fine-tuning, which motivates the work, is not followed through—no NLP experiments are provided.

2. The notation for sensitivity scores (e.g., $S_k^I$) is inconsistent and confusing, alternating between scalars and matrices.

3. The aggregation scheme in AFLA is heuristic and lacks theoretical justification or analysis.

4. Lastly, reliance on a clean proxy set may be limiting in privacy-sensitive applications.

---

> ### Author Rebuttal · Authors · 2025-07-31
>
> Dear Reviewer Egmb：
>
> Thank you for your thoughtful review and for raising key concerns regarding our work. We hope the following responses will address your concerns and update the score.
>
> **W1&Q1: Effectiveness on NLP tasks**
>
> To validate the generalizability of RFedLR, we conduct an additional experiment on NLP tasks. Following existing federated PEFT works [1][2][3], we perform an experiment on the MNLI task from the GLUE benchmark. We utilize Roberta-Base as the initial global model. The experimental setup mirrored our vision experiments, with clients holding non-IID data partitions and operating under a 20% symflip label noise. As shown in the Table R1, RFedLR achieves a accuracy of 71.42%, significantly outperforming all baseline methods. The results confirm that our framework is effective for NLP tasks in noisy federated settings.
>
> *Table R1: Comparison with state-of-the art methods on MNLI task from GLUE benchmark when the noise rate is 0.2 and noise type is symflip. We take the average test accuracy of the local models for demonstration.*
> | Methods | FedPF | FedBF | FedAP | FedLR | SLoRA | FFA-LoRA | RoLoRA | FlexLoRA | RFedLR    |
> |:-------:|:-----:|:-----:|:-----:|:-----:|:-----:|:-----:|:------:|:-----:|:------:|
> |   Acc  |  66.45 | 67.64 | 67.04 | 67.60 | 69.46 | 64.90    | 68.20  | 67.60    | **71.42** |
>
> Due to the time constraints of the rebuttal period, these NLP experiments were conducted for one representative task and noise setting. We will provide the full experimental results in the final version.
>
> **W2&Q2: Notational Clarity and Consistency**
>
> We thank the reviewer for pointing out this ambiguity in our notation. $S_k^i$ in Eq.5 is a scalar representing the sensitivity of a single parameter. In Eq.6, the collection of these sensitivity scores forms a sensitivity tensor $S_k$, which has the same dimensions as its corresponding parameter tensor. Therefore, $M_k$ in Eq.7 is a binary mask tensor with the same shape. We will clarify these symbols and their dimensions in the final version.
>
> **W3&Q3: Empirical validation of the AFLA weighting scheme**
>
> To provide the empirical evidence for our AFLA weighting scheme, we perform experiments in a controlled environment designed to validate its ability to mitigate noise amplification. We design a federated learning system with three clients. Client A is configured to be large but noisy (12,000 data points, 80% noise), Client B is configured to be small but clean (2,000 data points, clean), and Client C serves as a control (6,000 data points, 20% noise).
>
> As demonstrated in the Table R2, the standard FedAvg approach, which relies solely on data size, naively assigns the highest weight to the noisy Client A. This fixed weighting leads to severe noise amplification from Client A, resulting in a poor final accuracy of only 29.54%. In contrast, our AFLA mechanism considers both importance scores and data size to give more comprehensive weight allocation. It identifies Client A as an unreliable participant and reduces its average aggregation weight to 0.35. Simultaneously, it raises the weight of the small but reliable Client B to 0.32. This dynamic re-weighting mitigate the noise propagation and significantly improves accuracy.
>
> *Table R2: Empirical validation of the AFLA weighting scheme in a controlled 3-client scenario. The weights in AFLA  are dynamic each round, we show the average weight across all rounds to summarize its effective influence.*
> | Method |The Weighting Basis| Weight of Client A| Weight of Client B|Weight of Client C| Acc(%) |
> |:---:|:---:|:---:|:---:|:---:|:---:|
>  |FedAvg|Data Size|0.60 | 0.10 | 0.30 | 29.54 |
>  |AFLA|Importance & Data Size|0.35 | 0.32 | 0.33| 38.87 |
>
>
> **W4: Discussion of privacy restrictions**
>
> We agree that the reliance on a public proxy dataset can be a limitation. We have transparently acknowledged in the conclusion of our paper (Section 6). However, our method preserves the foundational privacy guarantee of federated learning, i.e., no private client data is ever shared. We follow the published federated learning research [4] in setting up proxy datasets, which can come from small public datasets relevant to the task, or from trusted participants in the federated learning system. In our work, we require only a mini-batch dataset to serve this purpose. Crucially, the proxy dataset does not need to be consistent with the local client data distribution, and the generated noise proxy dataset does not need to match locally specific noise patterns.
>
> **Q4: Clarification of Figure 2**
>
> We will clarify the caption of Figure 2 to indicate that “Federated LoRA” refers to the baseline FedLR method, where FedAvg is directly applied to LoRA. In addition, the accuracy difference between clean and noisy environments for federated FFT reaches 5.04%. However, the accuracy difference between Federated LoRA (FedLR) in clean and noisy environments reaches 11.48%. The presence of label noise poses a significant barrier to the robust performance of federated LoRA (FedLR). We will add specific values to Figure 2 in the final version.
>
> **Q5: Ablation experiments for the importance score**
>
> Fisher information (F) is used to approximate the importance of parameters on model decisions. Higher Fisher values indicate a greater influence of a parameter on the model output. The noise sensitivity score (S) measures the sensitivity of parameter updates to label noise. Lower sensitivity scores indicate greater stability and reliability. The formula $I = F / (S + \epsilon)$ aims to prioritize parameters that have a high impact on the task and are robust to label noise.
>
> We supplement this with ablation experiments (Table R3) to analyze the respective contributions of F and S to the final performance. We compare the RFedLR method ("Ours") with two variants: one that removes the Fisher information component ("w/o F," using $I = 1/ (S + \epsilon)$ ) and the other that removes the noise sensitivity component ("w/o S," using $I = F$).
>
> *Table R3: Ablation study for the hybrid importance score. The table compares the average test accuracy (%) of RFedLR (Ours) with variants that exclude Fisher information (w/o F) or noise sensitivity (w/o S) from the importance calculation in Eq. (10)*
> | Method | µ=0.2 (pairflip) | µ=0.2 (symflip) | µ=0.4 (pairflip) | µ=0.4 (symflip) |
> | :--- | :---: | :---: | :---: | :---: |
>  | Ours  | **83.12** | **86.97** | **67.08** | **84.64** |
>  | w/o F| 82.75 | 85.23 | 66.69 | 84.38 |
>  | w/o S| 77.42 | 80.86 | 61.04 | 77.44 |
>
> The significant performance drop in the "w/o S" case confirms that accounting for noise sensitivity is crucial for robust aggregation. Using Fisher information alone is insufficient and can be misleading in noisy environments. The full method always outperforms the “w/o F” case, which indicates that Fisher information further refines the importance score.
>
> **References**
>
> [1] Improving LoRA in Privacy-preserving Federated Learning, in ICLR 2024.
>
> [2] FedPETuning: When Federated Learning Meets the Parameter-Efficient Tuning Methods of Pre-trained Language Models, in ACL 2023.
>
> [3] FedPFT: Federated Proxy Fine-Tuning of Foundation Models, in IJCAI 2024.
>
> [4] Revisiting Weighted Aggregation in Federated Learning with Neural Networks, in ICML 2023.

---

> > ### Comment · Reviewer_Egmb · 2025-08-05
> > **Thanks for the rebuttal**
> >
> > Thanks for the detailed responses. All of my questions and concerns have been addressed. I would increase my points.

---

> > > ### Author Response · Authors · 2025-08-06
> > >
> > > Thank you for your valuable time and constructive comments. Your insightful feedback is instrumental in enhancing the quality of our paper.

---

### Official Review · Reviewer_yHga · 2025-06-29

**Clarity:** 3
**Significance:** 2
**Originality:** 2
**Rating:** 4
**Confidence:** 4

**Summary:**

The paper tackles the challenge of LoRA training on noisy labels under the Federated Learning scenario. The authors do so in four steps. First, the parameters most susceptible to model drifts due to noisy labels are measured by comparing differences between a clean proxy dataset and its noisy variant derived by injecting random label noise to it. Then a threshold is defined which allows to heuristically control the proportion of parameters (i.e. how much of the sensitive parameters) to drop out during local backward propagation, implemented via masking matrices. Thirdly, Fisher Information Matrix is used to compute the contribution importance of the decomposed LoRA parameter matrices for each of the clients' locally trained model from the previous step. Lastly, at aggregation time, the LoRA parameter weights to use for aggregation are chosen based on the importance values from the previous step. The proposed method is called RFedLR.

Experiments are performed with the Cifar100 and ImageNet 21k datasets on ViT models, and RFedLR is compared against other state-of-the-art and shown to significantly outperform them in the various training scenarios.

**Questions:**

1. Please describe, in detail, how/where a realistic proxy dataset and its subsequent noisy version may be derived from in order to make this solution more generalizable to practical scenarios.
2. How do we determine the value for Tau?
3. How and why where the FIM and the IM determined? I.e. what is the intuition behind their usage?
4. How does this perform across highly data heterogeneous environments?
5. Can this be adapted by other variations of LoRA such as AdaLora
6. What are the privacy implications of sharing the Parameter Importance matrices?

**Ethical Concerns:**

["NO or VERY MINOR ethics concerns only"]

**Final Justification:**

Concerns about applicability and generalizability still remain.

**Limitations:**

There may be privacy issues involved, as with all topics related to FL in general, but I do not foresee any problems specific to this paper.

**Paper Formatting Concerns:**

None.

**Quality:**

3

**Strengths And Weaknesses:**

Strengths -
1. The problem tackled is an interesting one, few papers explore this specific problem.
2. The various methods used for importance sampling at different stages is creatively applied.
3. The results are significantly better than state-of-the-art.

Weaknesses -
1. The use of a clean pre-available proxy data and adding noise artificially to determine importance is not a good practical solution since it is a very strong assumption that such data will always exist and that the noise is of a similar distribution to the clients' baseline data. How can that be guaranteed? This may be a very fundamental problem since every step of the solution depends on this.
2. The experimental section and discussion sections is lacking in diversity in terms of data heterogeneity (a fundamental concern for all FL systems) and systems (number of total clients, clients per round, datapoints per client).
3. The state-of-the-art compared against are not designed to tackle this specific challenge so the numbers may not be fair. The state-of-the-art are designed (mostly) for tackling LoRA challenges under data heterogeneity, which this paper completely avoids. On the other hand, state-of-the-art that are relevant (e.g. FedFixer) are not compared against.
4. While the individual steps are clearly described, how it all fits together is not.
5. Privacy implications are not discussed at all.

---

> ### Author Rebuttal · Authors · 2025-07-31
>
> Dear Reviewer yHga：
>
> Thank you for the valuable comments. We hope our responses address your concerns and provide a clearer understanding of our work.
>
> **W1&Q1: Practicality and assumption of the proxy dataset**
>
> We acknowledge that requiring a clean proxy dataset is a significant consideration, as we mentioned in Section 6. We follow the proxy dataset settings in published federated learning work [1] to achieve robust federated learning.
>
> In response to Q1, a proxy dataset can be sourced from small, publicly available datasets relevant to the task, or from a trusted, verifiable participant in FL systems. We agree that this is a constraint and will emphasize it in the limitations.
>
> In response to W1, the effectiveness of our method does not require that the distribution or noise pattern of the proxy dataset must be similar to the local data.
> 1. Robustness to Data Distribution Mismatch: In our setup, the proxy dataset is a uniformly distributed mini-batch randomly sampled from the CIFAR-100 test set. However, the local data was explicitly configured to be non-IID. Especially in the highly data heterogeneous environments (Refer to Q4), our method performs well, proving distribution matching is unnecessary.
> 2. Robustness to Noise Type Mismatch: The server injects uniform symmetric flip (symflip) noise into the proxy data for its analysis. However, our method achieves its most significant performance gains when clients are affected by structured pairflip noise (Table 2&3). This demonstrates that our approach is effective even with mismatched noise types.
>
> The proxy dataset is not used to simulate client local data, but as a common benchmark to identify parameters that are unstable to noise. This instability is a general property of model parameters, rather than their response to specific data distributions or noise patterns.
>
> **W2: Experimental diversity and system scale**
>
> To address this, we conduct experiments to simulate a more realistic, large-scale cross-device scenario. In this setup, we use 100 total clients with a 10% participation rate (10 selected per round). The results show RFedLR framework maintains its performance advantage in large-scale systems.
>
> *Table R1: Performance Comparison in a large-scale federated setting (100 total clients, 10% participation rate) with 40% pairflip noise. We take the average test accuracy for demonstration.*
> |Method|FedLR|SLoRA|FFA-LoRA|RoLoRA|FlexLoRA|RFedLR|
> |:-:|:-:|:-:|:-:|:-:|:-:|:-:|
> |Acc|59.04|53.91|58.74|58.75|54.17|**59.61**|
>
> Furthermore, to address the reviewer's concern about the diversity of data heterogeneity, we conduct experiments under a severe non-IID setting (the Dirichlet concentration parameter is 0.1), as detailed in our response to Q4 (Table R5). These results show that RFedLR is more effective in highly heterogeneous data distribution, demonstrating its robustness.
>
> These new results provide a more comprehensive validation of our method, confirming that RFedLR is effective in diverse federated systems.
>
> **W3: Comparison of SOTA federated noise learning methods**
>
> The experiments in Tables 2&3 focused on comparing RFedLR with other federated PEFT methods. As our work introduces a novel framework within the federated PEFT domain, we believe it is necessary to compare its performance with federated PEFT methods.
>
> We agree that comparing with the most relevant state-of-the-art is essential. We implement and evaluate robust federated learning methods, FedCorr [2], RHFL [3], FedFixer [4].
> As these methods are for conventional full-parameter FL, we adapt their core noise-handling mechanisms to our federated PEFT setting for a fair comparison. The experiments are performed under a noise rate of 0.4. These results (Table R2) demonstrate that RFedLR outperforms these robust FL methods. We analyze that these methods cannot mitigate the aggregation bias inherent in federated LoRA, while our framework is designed with SRT to counter local noise amplification and AFLA to address aggregation bias and noise propagation.
>
> *Table R2: Performance comparison of RFedLR with SOTA robust federated learning methods under a noise rate of 0.4. Values represent average test accuracy (%).*
> |Noise Type|FedCorr|RHFL|FedFixer|RFedLR|
> |:-:|:-:|:-:|:-:|:-:|
> |Pairflip|55.82|57.44|55.61|67.08|
> |Symflip|79.80|84.72|77.82|84.64|
>
> **W4: Integration of each components**
>
> The process for a single round of RFedLR is as follows:
>
> - Server-Side Preparation (AFLA): At the start of a round, the server selects the more important LoRA matrix (A or B) as trainable and distributes the global model to the participating clients.
> - Client-Side Local Tuning (SRT): Each client first uses the public proxy data to calculate a noise-sensitivity mask for the trainable LoRA matrix. During local training, this mask is used to selectively update only the most noise-sensitive parameters, enhancing local robustness.
> - Server-Side Aggregation (AFLA): After clients upload their updated matrices, the server calculates a hybrid aggregation weight for each client based on matrix importance and data size. It then performs a weighted average of these updates to create the new global model.
>
> To further clarify this workflow, we will add the detailed pseudocode to the appendix of our paper.
>
> **W5&Q6: Privacy implications**
>
> We would like to clarify a potential misunderstanding. Clients do not compute or share any “parameter importance matrices”. In each round, the client performs local training on the specified LoRA matrix and calculates an importance score. In the aggregation phase, each client only uploads an updated LoRA matrix and a scalar importance score. We believe that transmitting a scalar score computed on a public proxy dataset does not introduce substantial new privacy risks. We will discuss the privacy impact in the final version.
>
> **Q2: Sensitivity analysis for $\tau$**
>
> The parameter $\tau$ controls the proportion of parameters updated in our SRT mechanism. To provide best performance, we conduct a sensitivity analysis. Our results show that performance peaks at our chosen value of $\tau=0.2$. A smaller value overly constrains the model, while a larger value diminishes the regularization effect against noise.
>
> *Table R3: Parameter analysis for $\tau$ in the SRT module. The analysis is conducted under the 20% pairflip noise setting.*
> |$\tau$|0.1|0.2|0.3|0.4|0.5
> |:-:|:-:|:-:|:-:|:-:|:-:|
> |Acc|71.31|**83.12**|81.62|77.89|76.08|
>
> **Q3: The intuition behind FIM and IM**
>
> We adopt the empirical FIM to quantify the parameter importance, as it is a classic and effective measure of parameter impact on model prediction. Parameters with high Fisher Information values most affect predictive ability.
>
> The "IM" mentioned by the reviewer likely refers to our "hybrid importance score", denoted as $I$ in Eq.10, which combines FIM with our noise sensitivity metric. Although FIM can identify important parameters under ideal conditions, it ignores the parameter stability in noisy environments. Therefore, we design the hybrid importance score "$I = F / (S + \epsilon)$" to balance the contribution and reliability of parameters. We conduct ablation experiments on these two components (Fisher information and Noise sensitivity) to justify the design.
>
> To validate this design choice, our ablation study (Table R4) that empirically demonstrates the superiority of this hybrid score over using either Fisher Information or noise sensitivity alone. The results confirm that both components contribute positively, justifying our combined formulation.
>
> *Table R4: Ablation study for the hybrid importance score. The table compares the average test accuracy (%) of RFedLR (Ours) with variants that exclude Fisher information (w/o F) or noise sensitivity (w/o S) from the importance calculation in Eq.10*
> |Method|µ=0.2 (pairflip)|µ=0.2 (symflip)|µ=0.4 (pairflip)|µ=0.4 (symflip)|
> |:-:|:-:|:-:|:-:|:-:|
> |Ours|**83.12**|**86.97**|**67.08**|**84.64**|
> |w/o F|82.75|85.23|66.69|84.38|
> |w/o S|77.42|80.86|61.04|77.44|
>
> **Q4: Performance under highly data heterogeneous environments**
>
> We conduct the experiments under highly data heterogeneous environments by setting the Dirichlet concentration parameter to 0.1, with a noise rate of 0.4. The results (Table R5) show that the performance advantage of RFedLR becomes even more pronounced in this severe data heterogeneous scenario. We attribute this to the AFLA mechanism, which weights clients based on the amount of data, and the estimated importance and stability of their LoRA updates, ensuring robust aggregation results in highly non-IID scenarios.
>
> *Table R5: Comparison with the state-of-the-art methods under severe data heterogeneous scenario (Dirichlet concentration parameter is 0.1) when the noise rate is 0.4. We take the average test accuracy for demonstration.*
> |NoiseType|FedPF|FedBF|FedAP|FedLR|SLoRA|FFA-LoRA|RoLoRA|FlexLoRA|RFedLR|
> |:-:|:-:|:-:|:-:|:-:|:-:|:-:|:-:|:-:|:-:|
> |Pairflip|42.62|50.61|50.98|51.90|50.83|48.38|52.06|51.75|**62.56**|
> |Symflip|63.22|71.70|62.56|73.01|75.24|68.55|74.72|72.33|**80.48**|
>
> **Q5: Adaptation to LoRA variants**
>
> The core modules SRT and AFLA in RFedLR focus on the LoRA matrix parameters and do not rely on their fixed rank or structure. Therefore, the RFedLR framework can be effectively combined with other LoRA variants. For example, AdaLoRA dynamically allocates optimal, heterogeneous matrix ranks to different layers, and then RFedLR performs robust local training and global aggregation of these allocated matrices.
>
> **References**
>
> [1] Revisiting Weighted Aggregation in Federated Learning with Neural Networks, in ICML 2023.
>
> [2] Fedcorr: Multi-stage federated learning for label noise correction, in CVPR 2022.
>
> [3] Robust federated learning with noisy and heterogeneous clients, in CVPR 2022.
>
> [4] FedFixer: Mitigating heterogeneous label noise in federated learning, in AAAI 2024.

---

> > ### Comment · Reviewer_yHga · 2025-08-05
> > **Reviewer Response.**
> >
> > Thank you for your response and additional results.
> >
> > 1. I understand, the proposed system does not explicitly require similar proxy and data distributions, but it is highly impacted by it. The interplay between the model architecture, training setup and parameters, the proxy dataset and the label-flipping methods are important to detect the sensitivity. So the root question is - how close do these have to be in the proxy setting vs the application scenario? Do the authors have a bound, some heuristics, or intuition behind how to judge this? What would happen, for example, if the proxy dataset was CIFAR-100 (a 10 label subset with images most similar to CIFAR-10) and the FL system was CIFAR-10?
> >
> > 2. Thank you for all the new results and explanations, the effort is highly appreciated. There are still concerns such as lack of diversity of models/datasets/non-IIDness, as well as questions as to why the noise rates are different (0.4 vs 0.2) and why these specifically, but it is relatively convincing.
> >
> > 3. The need to tune tau is a concern. It is not clear if this needs to be done for every different task/scenario.
> >
> > The author responses somewhat alleviate the concerns. However, the reliance on proxy datasets, tau tuning and insufficient experiments variety are limiting.
> >
> > Since the authors do address the proxy issue in the limitations somewhat, and there is a time constraint for the range of experiments, I am increasing the score to 4. I can see it being useful in very specific situations, but the generalizability and applicability is still questionable.

---

> > > ### Author Response · Authors · 2025-08-05
> > > **Reply to reviewer yHga**
> > >
> > > Thank you for the thoughtful follow-up, the detailed engagement with our responses, and for raising the score. We understand that your remaining concerns primarily relate to the generalizability and applicability of our method. In this response, we offer further clarification and discussion on these points.
> > >
> > > **Q1. Robustness to proxy dataset mismatch**
> > >
> > > To validate the generalizability and adaptability of our proposed method, we supplement experiments by setting the proxy dataset to CIFAR-10, while the local dataset remain CIFAR-100. This setup introduces a significant mismatch in the label space and task complexity. This specific configuration is chosen to facilitate a timely comparison during the discussion period, as we already had baseline results on the CIFAR-100 local dataset.
> > >
> > > The results in Table R6 show that the performance of RFedLR is reduced when using the mismatched proxy (CIFAR-10) compared to using the matched proxy (CIFAR-100). However, **even with this significant mismatch, RFedLR remains robust**. RFedLR (CIFAR-10) still outperforms other state-of-the-art methods in various scenarios.
> > >
> > > *Table R6. Performance comparison with the state-of-the-art  methods, where the local dataset is set to a subset of CIFAR-100. "RFedLR(CIFAR-10)” and “RFedLR(CIFAR-100)” respectively indicate that we use a subset of CIFAR-10 and CIFAR-100 as the proxy dataset. The best and second-best results are highlighted in **bold**.*
> > > | Method | µ=0.2 (pairflip) | µ=0.2 (symflip) | µ=0.4 (pairflip) | µ=0.4 (symflip) |
> > > |:-:|:-:|:-:|:-:|:-:|
> > > | FedPF|67.98|75.64|50.10|72.79|
> > > | FedBF|74.78|83.32|55.95|78.50|
> > > | FedAP|75.48|82.81|55.60|78.62|
> > > | FedLR|75.28|83.65|56.62|79.69|
> > > | SLoRA|76.48|84.50|56.77|81.51|
> > > | FFA-LoRA|73.20|80.40|54.78|77.41|
> > > | RoLoRA|75.73|**84.55**|56.93|81.31|
> > > | FlexLoRA|74.68|83.61|56.58|79.71|
> > > | RFedLR (CIFAR-10)|**76.70**|84.54|**57.22**|**81.96**|
> > > | RFedLR (CIFAR-100)|**83.12**|**86.97**|**67.08**|**84.64**|
> > >
> > > The purpose of using the proxy dataset is to estimate the parameter sensitivity to gradient perturbations induced by label noise. While a task-relevant but non-identical proxy dataset may not perfectly capture parameter sensitivity, **it is still sufficient to detect the most significant instabilities**, thereby identifying the parameters most susceptible to noise-induced instability.
> > >
> > > **Q2. Concerns about experimental diversity**
> > >
> > > We appreciate your acknowledgment of our new experiments. We select noise rates of 0.2 and 0.4 to represent scenarios with varying noise intensities. This is a common practice in the field of label noise research [1][2][3] and allows for a clear demonstration of performance under different noisy levels.
> > >
> > > We acknowledge that validation on more datasets and models could further demonstrate generalizability. However, we believe the effectiveness of the RFedLR framework has been demonstrated through comprehensive experiments across a wide array of diverse and challenging scenarios. These evaluations include different **noise levels** (0.2 and 0.4) and **noise types** (pairflip and symflip), different **system scales** (5 and 100 clients), and multiple levels of **data heterogeneity**(Dirichlet parameter of 0.1 and 0.5). In addition, we provide the experimental results on NLP tasks to demonstrate the broad applicability of our approach (Refer to Table R1 in the response to Reviewer Egmb).
> > >
> > > **Q3. The issue of tuning $\tau$**
> > >
> > > We concede that fine-tuning for a specific task might yield improvements for peak performance, but this is not a prerequisite for the effectiveness of our method. As shown in Table R3, while performance peaks at $\tau=0.2$, it remains relatively stable within a reasonable range (0.2-0.3).
> > >
> > > In all experiments, we consistently use a fixed value of $\tau=0.2$ without any adjustments. The fact that our framework, using this fixed $\tau$, **achieve state-of-the-art performance across various challenging and diverse settings (i.e., different noise rates, noise types, system scales, and data heterogeneity levels) demonstrates the robustness of this parameter choice**.
> > >
> > > We will incorporate your feedback by enhancing the discussion on the generalization of our method, the rationale for choosing $\tau$, and the requirements for proxy datasets in the final version to make the paper more rigorous and comprehensive. We hope this further response can address your remaining concerns.
> > >
> > > [1] Co-teaching: Robust training of deep neural networks with extremely noisy labels, in NeurIPS 2018.
> > >
> > > [2] Unicon: Combating label noise through uniform selection and contrastive learning, in CVPR 2022.
> > >
> > > [3] Robust Training under Label Noise by Over-parameterization, in ICML 2022.

---

> > > > ### Comment · Reviewer_yHga · 2025-08-09
> > > > **Response**
> > > >
> > > > Most of my concerns have been addressed. I will increase my score.

---

> > > > > ### Author Response · Authors · 2025-08-09
> > > > >
> > > > > Thank you for your thoughtful feedback and for raising the score for our work. We appreciate your time and the constructive comments, which have helped improve the paper.

---

### Official Review · Reviewer_N5Xd · 2025-06-30

**Clarity:** 3
**Significance:** 2
**Originality:** 3
**Rating:** 4
**Confidence:** 4

**Summary:**

This work proposes a new federated PEFT framework, RFedLR, designed to improve global model performance when local clients possess label-noisy private datasets. RFedLR assumes the availability of a clean proxy dataset collected by the server, from which a synthetic label-noisy dataset is constructed. Both the clean and noisy datasets are distributed to the clients, who then compute the sensitivity of their local models by measuring the gradient discrepancy between models trained on the clean and noisy versions. Parameters identified as highly sensitive to label noise are deemed less reliable and are subsequently frozen during backpropagation. Furthermore, RFedLR incorporates an adaptive LoRA aggregation technique that accounts for both the data scale and parameter importance of each client, with the latter estimated using the Fisher Information Matrix. Experiments on the CIFAR-100 dataset demonstrate the effectiveness of the proposed RFedLR framework.

**Questions:**

1. What is the performance of clients who hold clean datasets? This is an important baseline that should be included, as it represents the theoretical upper bound of model performance.

2. How does RFedLR perform in federated settings with greater data heterogeneity? For instance, if the Dirichlet concentration parameter $\beta$ is increased to 0.1, does RFedLR lose its effectiveness? Furthermore, in scenarios where local data distributions are highly non-IID, is the small proxy dataset sufficient for accurately computing parameter sensitivity?

**Ethical Concerns:**

["NO or VERY MINOR ethics concerns only"]

**Final Justification:**

The authors address most of my concerns in the rebuttal. As a result, I increased my score.

**Limitations:**

Yes

**Quality:**

2

**Strengths And Weaknesses:**

**Strengths:**

1. The paper is well-written, and the proposed methodology is clearly presented and easy to follow.
2. The motivation behind RFedLR is promising. Label noise is a naturally occurring issue in real-world datasets, and addressing it in the context of FL is both important and timely.
3. The authors conduct several experiments to demonstrate the effectiveness of the proposed RFedLR framework.

**Weaknesses:**

1. The technical novelty of the proposed approach appears somewhat limited. Leveraging gradient discrepancy to estimate parameter sensitivity to label noise is not a new idea and has been explored previously, particularly in literature related to backdoor attacks.
2. The experimental evaluation is narrow in scope. The study is restricted to the CIFAR-100 dataset, which is relatively small and stylized. It would strengthen the paper to include experiments on larger-scale datasets or consider the applicability of RFedLR to LLM fine-tuning tasks, as done in recent works.
3. The paper lacks sufficient details on how the server constructs the synthetic noisy dataset. For instance, what process is used to inject noise? The experiments mention two types of label noise, but it is unclear whether the server-generated noisy dataset adheres to these same noise patterns.
4. A related concern is that the server is unaware of the actual data distribution or noise type on each client. What happens if the server assumes pairwise flip noise while some clients actually follow symmetric label noise? This mismatch may undermine the effectiveness of the gradient discrepancy analysis.
5. The evaluated FL setting is highly cross-silo, involving only five clients. To validate the generality and scalability of RFedLR, it is important to evaluate it under cross-device settings with larger numbers of clients.

---

> ### Author Rebuttal · Authors · 2025-07-31
>
> Dear Reviewer N5Xd：
>
> Thank you for your insightful review and valuable feedback. We address your key concerns in detail below, aiming to clarify and demonstrate the effectiveness of our proposed approach.
>
> **W1: The Novelty of the paper**
>
> In this paper, we investigate the novel problem of achieving robust federated PEFT in noisy environments. With the rapid growth of large-scale pre-trained models, this is a realistic and challenging problem. Compared to conventional robust federated learning, federated PEFT in noisy environments presents additional challenges. The low-rank factorization structure of LoRA can lead to noise amplification in noisy environments. Noisy perturbations generate new bias terms through matrix multiplication. Furthermore, federated LoRA aggregation suffers from inherent aggregation bias, exacerbated by noisy local updates, and this bias is propagated and amplified throughout the federated system.
>
> To address these challenges, we propose RFedLR framework, which consists of two modules: (1) SRT selectively updates noise-sensitive parameters to enhance robustness at the local fine-tuning phase, and (2) AFLA introduces an adaptive aggregation scheme at the collaborative update phase to mitigate aggregation discrepancies and curtail noise propagation.
>
> Prior works, particularly in backdoor defense, have leverage gradient analysis to detect parameters affected by  the backdoor trigger. The backdoor literature aims to detect deterministic signals from a single static attack. In contrast, our approach aims to measure the parameter stability against random label noise. Its goal is not to defend against targeted attacks, but rather at achieving generalized robust regularization to prevent overfitting. Besides, some work prunes or permanently disables outlier parameters. Unlike these approaches, SRT maintains all parameters active during forward propagation to ensure the model representability is not compromised, while selectively updating noise-sensitive parameters during backpropagation. This conditional update mechanism is a more sophisticated robust training mechanism. Furthermore, the parameter sensitivity score is combined with Fisher information to form a hybrid importance score, based on which aggregate weights are assigned to the clients.
>
> In summary, the novelty of our work lies in studying a new problem, analyzing its unique challenges, and proposing a new framework to address these challenges.
>
> **W2: Expansion of the experimental scope**
>
> As our paper focuses on the challenges of fine-tuning large-scale Pre-trained Models (PTMs), we choose ViT-B (pre-trained on ImageNet-21K) as our backbone. Its fine-tuning on CIFAR-100 follows a standard and challenging benchmark from recent visual PEFT literature [1], ensuring our baseline comparisons are meaningful.
>
> To demonstrate the broad applicability of our framework, we conduct experiments on a NLP fine-tuning task involving a language model (Roberta-Base model). We evaluate the performance of RFedLR on the MNLI task from GLUE benchmark with 20% symflip noise in a non-IID federated setting. The results are as follows:
>
> *Table R1: Comparison with the state-of-the art methods on MNLI task from GLUE benchmark when the noise rate is 0.2 and noise type is symflip. We take the average test accuracy of the local models for demonstration.*
> |Methods|FedPF|FedBF|FedAP|FedLR|SLoRA|FFA-LoRA|RoLoRA|FlexLoRA|RFedLR|
> |:-:|:-:|:-:|:-:|:-:|:-:|:-:|:-:|:-:|:-:|
> |Acc|66.45|67.64|67.04|67.60|69.46|64.90|68.20|67.60|**71.42**|
>
> The results show that RFedLR significantly outperforms existing federated PEFT methods in the language domain, confirming its applicability beyond image classification. We consider extending it to the LLM fine-tuning scenario in future work.
>
> **W3: The construction of the noisy proxy dataset**
>
> To clarify, the noisy proxy dataset $D_n$ is generated by injecting symflip noise into the clean dataset $D_c$, which randomly flips a label to any other class with uniform probability. Symflip noise represents a general form of label noise that does not rely on any assumptions about class similarity. By evaluating parameter sensitivity against this high-entropy noise, we can identify parameters that are broadly unstable to any incorrect supervision signal. This ensures that our sensitivity analysis is robust and not biased towards a specific, structured noise pattern. The approach allows RFedLR to learn a more general measure of noise sensitivity, enhancing the overall robustness.
>
> The design choice is validated by our experimental results (Tables 2&3), where RFedLR demonstrates superior performance under both symflip and pairflip noise settings, confirming the general applicability of the learned sensitivity measure. We will clarify this implementation detail in Section 4.1 of the revised manuscript.
>
> **W4: The construction of the noisy proxy dataset**
>
> The effectiveness of our method does not depend on the assumption that the distribution or noise pattern of the proxy dataset must be match to the local data.
>
> 1. Robustness to Data Distribution Mismatch: In our setup, the proxy dataset is a uniformly distributed mini-batch randomly sampled from the CIFAR-100 test set. However, the local data was explicitly configured to be non-IID. As demonstrated by our results, particularly in the highly data heterogeneous environments (Refer to Q2), our method performs well, which proves that distribution matching is unnecessary.
> 2. Robustness to Noise Type Mismatch: As clarified in our response to W3, our framework utilizes symflip noise to generate noisy proxy data. Table 2&3 show that RFedLR still demonstrates remarkable effectiveness when clients have structured pairflip noise. Besides, Table 1 shows that the introduction of SRT module results in a significant accuracy improvement under pairflip noise with different noise rates. This shows that a mismatch between the proxy noise and the local true noise does not undermine the effectiveness of our gradient discrepancy analysis.
>
> The proxy dataset is not used to simulate client local data, but to serve as a common, objective benchmark for identifying parameters that are unstable to noise. This instability is a general property of model parameters, rather than their response to specific data distributions or noise patterns.
>
> **W5: Effectiveness under cross-device settings**
>
> We agree that validating the scalability and generality of our method in a setting with a larger number of clients is essential for demonstrating its practical applicability. To address this concern, we conduct experiment in a large-scale setting designed to simulate a cross-device federated learning scenario. This experiment involves 100 total clients with a 10% participation rate, meaning 10 clients are sampled for training and aggregation in each communication round. The results, presented in the table below, show that RFedLR outperforms other methods in this more challenging and realistic environment.
>
> *Table R1: Performance Comparison in a large-scale federated setting (100 total clients, 10% participation rate) with 40% pairflip noise. We take the average test accuracy of the local models for demonstration.*
> |Method|FedLR|SLoRA|FFA-LoRA|RoLoRA|FlexLoRA|RFedLR|
> |:-:|:-:|:-:|:-:|:-:|:-:|:-:|
> |Acc|59.04|53.91|58.74|58.75|54.17|**59.61**|
>
> This result demonstrates that the effectiveness of our proposed RFedLR framework is not limited to small-scale, cross-silo settings but scales successfully to systems with more clients.
>
> **Q1: Performance of clean clients**
>
> We would like to clarify that this baseline is included in our analysis, presented in Figure 2 on page 2. The purpose of this figure is to visually demonstrate the gap in model performance between practical noisy the ideal clean scenarios. Specifically, Figure 2(b) shows the performance of Federated LoRA (FedLR), the baseline setting of our study. As can be seen from the figure, when all clients have clean datasets (shown by the gray dashed line), the final average accuracy of the model can reach 86.76%. When the client dataset contains noise (shown by the orange solid line), the accuracy drops to 75.28%. This 11.48% performance gap reveals the significant negative impact of label noise on Federated PEFT. This significant gap emphasizes the need to study a robust federated PEFT framework. Our proposed method RFedLR achieves an accuracy of 83.12% under the same noise conditions of the Figure 2. For clarity, we will add specific values to Figure 2 in the final version.
>
> **Q2: Performance under severe data heterogeneity**
>
> We have conduct the experiments in a highly non-IID setting by setting the Dirichlet concentration parameter to 0.1, with a noise rate of 0.4. The results (Table R3) show that the performance advantage of RFedLR becomes even more pronounced in this severe data heterogeneous scenario. We analyze the advantages to be attributed to the AFLA mechanism. AFLA weights clients based on the amount of data, and the estimated importance and stability of their LoRA updates, ensuring robust aggregation results in highly non-IID scenarios.
>
> *Table R3: Comparison with the state-of-the art method under severe data heterogeneous scenario (Dirichlet concentration parameter is 0.1) when the noise rate is 0.4. We take the average test accuracy of the local models for demonstration.*
> |NoiseType|FedPF|FedBF|FedAP|FedLR|SLoRA|FFA-LoRA|RoLoRA|FlexLoRA|RFedLR|
> |:-:|:-:|:-:|:-:|:-:|:-:|:-:|:-:|:-:|:-:|
> |Pairflip|42.62|50.61|50.98|51.90|50.83|48.38|52.06|51.75|**62.56**|
> |Symflip|63.22|71.70|62.56|73.01|75.24|68.55|74.72|72.33|**80.48**|
>
> The small proxy dataset serves as a general benchmark for identifying which parameters are inherently unstable when exposed to label noise. This parameter-level property is largely independent of the specific data distribution.
>
> **References**
>
> [1] Sensitivity-aware visual parameter-efficient fine-tuning, in ICCV 2023.

---

> > ### Comment · Reviewer_N5Xd · 2025-08-03
> > **Response to rebuttal**
> >
> > Thank you for your response and for conducting the additional experiments. While some of my concerns have been addressed, several critical issues remain, as also noted by other reviewers. For instance, Reviewer Egmb highlights that the evaluation is limited to the simple and stylish CIFAR-100 dataset, which significantly restricts the generalizability of the findings. Additionally, concerns have been raised regarding the privacy implications of requiring a proxy dataset on the server side.

---

> > > ### Author Response · Authors · 2025-08-04
> > > **Reply to reviewer N5Xd**
> > >
> > > Thank you for the thoughtful follow-up and for raising these critical points. Your main concerns are about the selection of evaluation datasets and the sharing of proxy datasets. Here, we provide more in-depth clarification and discussion on these two issues.
> > >
> > > Verifying the effectiveness of methods on diverse and large-scale datasets is key to evaluating their generalization ability. We chose CIFAR-100, the most commonly used benchmark in FL work [1][2][3], as the main benchmark for visual tasks to ensure fair comparison **with previous work and reproducibility of results**. To address the limitations of a single domain as suggested, we have supplemented the experimental results on NLP tasks (**MNLI task of GLUE benchmark, which contains 393k training samples and 20k test samples**) [4][5] in the previous round of responses. It shows that RFedLR also significantly outperforms all baseline methods in the language domain, demonstrating its generalization. Extending this framework to larger-scale vision and language models will be the focus of our future work.
> > >
> > > Regarding concerns about the potential privacy risks associated with the introduction of proxy datasets. In many state-of-the-art works on FL [6][7][8][9], introducing a proxy dataset to assist the server is a common technical solution. Our work initially follows the setting of existing work and further minimizes potential privacy risks in design: Our method only requires **a proxy dataset of "mini-batch" size** (set to 256 in the experiment), which is much smaller than the local dataset and accounts for approximately 0.5% (256/50,000) of the total sample size of the local data. In addition, **the proxy dataset does not need to be completely consistent with the private data**. As we demonstrate (Table 2&3&R3), our method works well even if the distribution and noise type of the proxy data do not match the local data. This shows that the proxy data is used to explore the inherent parameter properties of the model, rather than the client data itself. Crucially, the server only uses it to calculate gradients to estimate parameter sensitivity, and **does not use it to train the global model**. Therefore, we contend that the minimal and well-controlled privacy consideration of using a proxy dataset is a justifiable trade-off for the significant enhancement in robustness of federated PEFT.
> > >
> > > We hope this further clarification addresses your concerns.
> > >
> > > [1] FedDiv: Collaborative noise filtering for federated learning with noisy labels, AAAI 2024.
> > >
> > > [2] Fedcorr: Multi-stage federated learning for label noise correction, in CVPR 2022.
> > >
> > > [3] Robust federated learning with noisy and heterogeneous clients, in CVPR 2022.
> > >
> > > [4] GLUE: A Multi-Task Benchmark and Analysis Platform for Natural Language Understanding, in ICLR 2019.
> > >
> > > [5] FedPETuning: When Federated Learning Meets the Parameter-Efficient Tuning Methods of Pre-trained Language Models, in ACL 2023.
> > >
> > > [6] Revisiting Weighted Aggregation in Federated Learning with Neural Networks, in ICML 2023.
> > >
> > > [7] CDKT-FL: cross-device knowledge transfer using proxy dataset in federated learning, in EAAI 2022.
> > >
> > > [8] Sageflow: Robust Federated Learning against Both Stragglers and Adversaries, in NeurIPS 2021.
> > >
> > > [9] Fltrust: Byzantine-robust federated learning via trust bootstrapping, in NDSS 2021.

---

> > > > ### Comment · Reviewer_N5Xd · 2025-08-05
> > > > **Response**
> > > >
> > > > Thanks for your further clarification. I will increase my score to borderline accept.

---

> > > > > ### Author Response · Authors · 2025-08-06
> > > > >
> > > > > Thank you for taking the time to review our responses. We appreciate your insightful feedback, which will significantly improve the quality of the paper.

---

### Official Review · Reviewer_MLqt · 2025-07-02

**Clarity:** 3
**Significance:** 3
**Originality:** 3
**Rating:** 5
**Confidence:** 4

**Summary:**

This paper is the first to study the problem of robust federated PEFT in label noisy scenarios. LoRA is a common PEFT method that helps reduce costs but is highly sensitive to label noise and aggregation inconsistency in the federated setting. RFedLR addresses this problem by introducing two main components: sensitivity-aware robust tuning, which identifies and updates only the parameters most sensitive to noise during local tuning, and adaptive federated LoRA aggregation, which selectively aggregates LoRA updates based on their importance and stability. The paper evaluates RFedLR in different label noisy scenarios and shows that it significantly outperforms existing methods in accuracy and efficiency.

**Questions:**

The authors should provide additional resource consumption comparisons with other methods and sensitivity analysis experiments on all hyperparameters. Besides, the authors should correct the inconsistencies and typos in the paper (mentioned in weaknesses), especially the discrepancy between Eq.5 and its description.

**Ethical Concerns:**

["NO or VERY MINOR ethics concerns only"]

**Final Justification:**

The authors’ rebuttal, particularly the new experiments, has resolved my initial concerns about the evaluation. This is a valuable contribution to federated graph learning, and I now recommend acceptance.

**Limitations:**

Yes

**Quality:**

3

**Strengths And Weaknesses:**

Strengths

1. This paper explores the practical but underexplored problem of robust federated fine-tuning in noisy scenarios, providing a clear motivation for this research.

2. The proposed method is well-designed and innovative. Clear illustrations and figures make the method easy to understand.

3. The experimental setup is thorough and provides a convincing evaluation, including ablation studies, comparisons under different noise types and levels, and performance across clients.

Weaknesses

1. The calculation of gradient differences and Fisher information in RFedLR will introduce additional overhead. This may affect its adaptability on low-resource devices. How does the computation time or resource consumption of this method compare with the baseline method?

2. The parameter keep rate \tau in SRT is fixed to 0.2, and the balance hyperparameter \lambda in AFLA is fixed to 0.4. There is a lack of discussion or sensitivity analysis on their selection, which may affect the generalizability of the method.

3. Typos.
- On page 5, in Eq. 5, the authors define noise sensitivity as the subtraction of two gradients, but previously described as "the magnitude of the gradient discrepancy". There is a difference between these two statements. Is the intention here to calculate the absolute value or norm of the difference?

- For formula 10, it is recommended to specify whether the original sensitivity score or the normalized score is used.

- The numerical format of Table 3 is inconsistent, please check.

- There is a typo in line 171 on page 4, which should be "the A and B matrices".

---

> ### Author Rebuttal · Authors · 2025-07-31
>
> Dear Reviewer MLqt：
>
> Thank you for your valuable feedback and for your time reviewing our work. We hope our responses below help clarify the issues.
>
> **W1: Computational Overhead**
>
> We conduct a detailed analysis of the resource consumption of our proposed RFedLR compared to other baseline methods. As shown in Table R1, the results show that the additional memory cost of RFedLR is negligible. Specifically, its memory usage is 33642.74 MB, which is comparable to other baseline methods. Furthermore, RFedLR utilizes the fewest trainable parameters (0.1754%) and achieves the highest test accuracy among all compared methods.
>
> We will add a table summarizing these findings to the appendix and discuss these results in the main paper.
>
> *Table R1: Comparison of trainable parameter(%) relative to FFT and memory cost(MB) for various methods.*
> | Methods  | Trainable Parameters (%) | Memory Cost (MB) |
> |:--------:|:-----------------------:|:----------------:|
> | FedPF    | 0.3042                  | 32475.57         |
> | FedBF    | 0.2085                  | 33005.22         |
> | FedAP    | 1.1413                  | 33107.69         |
> | FedLR    | 0.6107                  | 33633.73         |
> | SLoRA    | 0.6107                  | 46973.41         |
> | FFA-LoRA | 0.5221                  | 31858.00         |
> | RoLoRA   | 0.5249                  | 34144.27         |
> | FlexLoRA | 0.6107                  | 32713.07         |
> | RFedLR   | 0.1754                  | 33642.74         |
>
> **W2: Hyperparameter Sensitivity**
>
> We include a sensitivity analysis for the key hyperparameters $\tau$ and $\lambda$.
> For the parameter keep rate $\tau$ in SRT (Table R2), our analysis shows that performance peaks at our chosen value of $\tau=0.2$, achieving an accuracy of 83.12%. When $\tau$ is too low, the model plasticity is limited. When $\tau$ is too high, the regularization effect of the selective update strategy is weakened and performance decline.
>
> *Table R2: Parameter analysis for $\tau$ in the SRT module. The analysis is conducted under the 20% pairflip noise setting.*
> | $\tau$ | 0.1| 0.2| 0.3|0.4|0.5
> |:----:|:-----:|:-----:|:-----:|:-----:|:-----:|
> | Acc  | 71.31 | **83.12** | 81.62 | 77.89 | 76.08 |
>
> For the balancing hyperparameter $\lambda$ in AFLA (Table R3), our results show that the performance of RFedLR is relatively stable for the selection of $\lambda$. When $\lambda$ is 0.4, RFedLR achieves the highest test accuracy, which can effectively balance the contribution of data scale and matrix importance.
>
> *Table R3: Parameter analysis for $\lambda$ in the AFLA module. The analysis is conducted under the 20% symflip noise setting.*
> | $\lambda$ | 0.1 | 0.2 | 0.3 | 0.4 | 0.5 | 0.6 | 0.7 | 0.8
> |:------:|:-----:|:-----:|:-----:|:-----:|:-----:|:-----:|:-----:|:-----:|
> | Acc   | 86.22 | 85.50 | 85.46 | **86.97** | 85.34 | 85.84 | 85.42 | 86.06 |
>
> We will include these results and corresponding charts in the paper to improve the integrity of the paper.
>
> **W3: Manuscript Clarifications and Corrections**
>
> We are grateful to the reviewer for the meticulous reading and for pointing out several areas for improvement in the manuscript.
>
> We will correct all identified typos and inconsistencies. Specifically, we will revise the description of Eq. 5 and update the formula to $S_{k}^{i} = \left| \nabla_{W_{k}^{i}}L_{c} - \nabla_{W_{k}^{i}}L_{n} \right|$ to accurately reflect the calculation of the magnitude of the gradient discrepancy. For Eq.10, we need to clarify that the unnormalized sensitivity score ($S_k^i$) is used in the calculation of the hybrid importance score to avoid ambiguity. We will correct the inconsistent numerical formatting in Table 3 and the typos in the final version.

---

> > ### Comment · Reviewer_MLqt · 2025-08-08
> > **Thanks for your response**
> >
> > In this rebuttal, the authors provide a detailed comparative analysis of computational costs, an analysis of hyperparameter sensitivity, and clarifications of unclear descriptions. Their commitment to incorporate these changes into the final manuscript will significantly improve its quality. With the improvements provided in their response, my concerns have been adequately addressed. I will update my score accordingly.

---

> > > ### Author Response · Authors · 2025-08-09
> > >
> > > Thank you for the insightful and constructive feedback, which has helped us improve our work. We also appreciate your acknowledgment and positive comments on our paper.

---

### Note · Authors · 2025-08-14

We sincerely thank all the reviewers for their valuable feedback and the AC for their diligent coordination throughout the review process.

During the rebuttal and discussion phase, we have carefully addressed all the reviewers' comments, providing additional diverse experimental results and clarifying potential misunderstandings. After discussion, most reviewers acknowledge the importance of our research problem, the novelty of our framework, and the comprehensive experimental validation.

We will incorporate all constructive suggestions into the final version to further enhance the quality of our paper.

---

### Decision · Program_Chairs · 2025-09-17

**Decision:**

Accept (poster)

**Comment:**

This paper studies an important problem of robust federated lLoRA against label noises. The key idea is to introduce clean proxy datasets, train the models on both the clean and noisy datasets over the clients, compare their the gradient discrepancy to keep/ignore the parameters sensitive/insensitive to the noises. The reviewers point out several unsolved issues of the current version including the reliance on proxy datasets may raise serious privacy issues and limit its generalizability, tau tuning can lead to non-trivial performance drop, and insufficient experimental evaluation. I recommend the authors to take the reviewers' suggestions into account to further improve their work.